REPORT

# Parallel phospholipid transfer by Vps13 and Atg2 determines autophagosome biogenesis dynamics

Rahel Dabrowski[1], Susanna Tulli[1], and Martin Graef[1,2]

During autophagy, rapid membrane assembly expands small phagophores into large double-membrane autophagosomes. Theoretical modeling predicts that the majority of autophagosomal phospholipids are derived from highly efficient non-vesicular phospholipid transfer (PLT) across phagophore–ER contacts (PERCS). Currently, the phagophore–ER tether Atg2 is the only PLT protein known to drive phagophore expansion in vivo. Here, our quantitative live-cell imaging analysis reveals a poor correlation between the duration and size of forming autophagosomes and the number of Atg2 molecules at PERCS of starving yeast cells. Strikingly, we find that Atg2-mediated PLT is non-rate limiting for autophagosome biogenesis because membrane tether and the PLT protein Vps13 localizes to the rim and promotes the expansion of phagophores in parallel with Atg2. In the absence of Vps13, the number of Atg2 molecules at PERCS determines the duration and size of forming autophagosomes with an apparent in vivo transfer rate of ~200 phospholipids per Atg2 molecule and second. We propose that conserved PLT proteins cooperate in channeling phospholipids across organelle contact sites for non-rate-limiting membrane assembly during autophagosome biogenesis.

## Introduction

Macroautophagy (hereafter "autophagy") is a conserved catabolic process with essential functions for cellular homeostasis and broad implications for disease and aging (Dikic and Elazar, 2018; Hansen et al., 2018; Leidal et al., 2018). Autophagy is characterized by the de novo formation of transient double-membrane organelles, termed autophagosomes, which enable cells to degrade an unparalleled scope of substrates in vacuoles or lysosomes. During autophagosome biogenesis, small membrane seeds nucleate from Atg9 and COPII vesicles derived from the Golgi/recycling endosomes and the (ER/ER–Golgi intermediate compartment, respectively (Davis et al., 2016; Ge et al., 2013; Ge et al., 2017; Ge et al., 2014; Karanasios et al., 2016; Kumar et al., 2021; Longatti et al., 2012; Mari et al., 2010; Puri et al., 2018; Shima et al., 2019; Yamamoto et al., 2012; Young et al., 2006; Zoppino et al., 2010). After nucleation, membrane seeds rapidly expand into large cup-shaped phagophores (or isolation membranes) around their cargo within minutes (Axe et al., 2008; Schütter et al., 2020; Tsuboyama et al., 2016; Xie et al., 2008). Upon phagophore closure, cargo is enclosed by the inner and outer membranes of resulting autophagosomes. Outer membranes fuse with vacuoles or lysosomes resulting in the degradation of the inner vesicles and cargo. Autophagosome formation is driven by conserved core autophagy protein machinery whose hierarchical assembly and function

have been characterized in molecular detail culminating in in vitro reconstitution of the process up to membrane expansion (Nakatogawa, 2020; Sawa-Makarska et al., 2020). In contrast to other membrane-bound organelles, cup-shaped phagophores and autophagosome are characterized by low protein content within their membranes and a very narrow intermembrane distance between closely apposed phospholipid bilayers (Bieber et al., 2022; Hayashi-Nishino et al., 2009; Ylä-Anttila et al., 2009). These unique features strongly suggest the existence of specialized molecular mechanisms underlying the expansion of forming autophagosomes.

Autophagosomes form in close spatial association with ER (Axe et al., 2008; Biazik et al., 2015; Bieber et al., 2022; Gómez-Sánchez et al., 2018; Graef et al., 2013; Hayashi-Nishino et al., 2009; Suzuki et al., 2013; Ylä-Anttila et al., 2009). During expansion, cup-shaped phagophores adopt a defined orientation, in which the rim contacts the ER and the convex backside attaches to the vacuole in yeast (Bieber et al., 2022; Graef et al., 2013; Hollenstein et al., 2019; Suzuki et al., 2013). The conserved autophagy protein Atg2 functions as a membrane tether at phagophore–ER contact sites (PERCS) and binds to the rim of the expanding phagophore via a C-terminal α-helix and coincidence binding of phosphatidylinositol-3-phosphate in complex with

[1]Max Planck Research Group of Autophagy and Cellular Ageing, Max Planck Institute for Biology of Ageing, Cologne, Germany; [2]Department of Molecular Biology and Genetics, Cornell University, Ithaca, NY, USA.

Correspondence to Martin Graef: martin.graef@cornell.edu.



the PROPPIN Atg18 (Chowdhury et al., 2018; Gómez-Sánchez et al., 2018; Graef et al., 2013; Kotani et al., 2018; Obara et al., 2008; Suzuki et al., 2013; Tamura et al., 2017; Zheng et al., 2017). Atg2 is a member of a phospholipid transfer protein family, which forms rod- or bridge-like protein structures with extended hydrophobic grooves (Melia and Reinisch, 2022; Neuman et al., 2022). Importantly, these hydrophobic grooves display non-vesicular phospholipid transfer (PLT) activity catalyzing phospholipid exchange between tethered membranes in vitro (Maeda et al., 2019; Osawa et al., 2019; Valverde et al., 2019). At the phagophore rim, Atg2 physically interacts with the transmembrane protein Atg9 (Gómez-Sánchez et al., 2018; van Vliet et al., 2022). Atg9 forms a homotrimeric complex with an inherent phospholipid scramblase activity (Ghanbarpour et al., 2021; Guardia et al., 2020; Maeda et al., 2020; Matoba et al., 2020; Orii et al., 2021). Mammalian ATG2A and ATG9A form a heteromeric complex, in which the opening of the hydrophobic groove of ATG2A aligns with an internal channel of ATG9A implicated in phospholipid scrambling (van Vliet et al., 2022). This spatial and functional arrangement suggests that Atg2 and Atg9 may constitute a minimal system for non-vesicular transfer and scrambling of phospholipids from the ER into the outer leaflet and equilibration between both leaflets of the phagophore membrane to allow for the formation of large autophagosomes. This model, however, does not explain how cells achieve the required net transfer of phospholipids into the phagophore because both Atg2- and Atg9-mediated activities are energy-independent and principally operate in a bidirectional manner (Ghanbarpour et al., 2021; Maeda et al., 2019; Maeda et al., 2020; Matoba et al., 2020; Osawa et al., 2019; Valverde et al., 2019). Notably, ATP-dependent fatty acid activation drives localized phospholipid synthesis within the ER, which promotes preferentially the incorporation of phospholipids into phagophore membranes and is required for efficient and productive expansion of forming autophagosomes (Orii et al., 2021; Schütter et al., 2020). Thus, localized phospholipid synthesis in the ER may be a driver of unidirectional phospholipid transfer across PERCS. Taken together, the cooperation of localized phospholipid synthesis, non-vesicular transfer, and scrambling provides a mechanistic model for phagophore expansion during autophagosome biogenesis.

In this study, we examined the quantitative and dynamic features of the current model for autophagosome biogenesis. Using quantitative live-cell imaging in yeast, we analyzed whether Atg2 clusters at PERCS can provide sufficiently high PLT to drive the expansion of forming autophagosomes at rates observed in vivo. Strikingly, we find Atg2 cooperates with the conserved bridge-like PLT protein Vps13 at the phagophore rim to ensure non-rate-limiting membrane assembly during autophagosome biogenesis.

## Results and discussion

Membrane assembly during autophagosome biogenesis depends on vesicular and non-vesicular phospholipid transfer (PLT; Fig. 1 A), but their relative quantitative contribution to the formation of autophagosomes has been unknown. For a first estimate, we

started with a theoretical model for autophagosome biogenesis (Fig. 1, A–C). Recent structural analyses suggest a narrow intermembrane distance of ~5 nm between the outer and inner membranes of mature autophagosomes in yeast (Fig. 1 A; Bieber et al., 2022). This key parameter enabled us to approximate the number of phospholipids and corresponding intermembrane volume of autophagosomes within the observed size range in cells as described in Materials and methods (Fig. 1 B). Assuming intermembrane volume is exclusively derived from the fusion of Atg9 (60 nm) and COPII (80 nm) vesicles (Shima et al., 2019; Yamamoto et al., 2012), we deduced non-vesicular PLT contributes ~75% of the 4 to 18 million phospholipids within autophagosome membranes across different sizes, consistent with recent estimates (Fig. 1, B and C; Bieber et al., 2022). Given that autophagosomes form within a few minutes (Schütter et al., 2020; Tsuboyama et al., 2016; Xie et al., 2008), these numbers highlight the need for highly efficient non-vesicular PLT mechanisms between expanding phagophores and ER (and potentially other organelles). Purified Atg2 can channel phospholipids between membranes in vitro (Maeda et al., 2019; Osawa et al., 2019; Valverde et al., 2019). However, it has been unclear whether Atg2 constitutes the only PLT protein at phagophores and, if so, whether its inherent PLT capacity is sufficient to expand phagophores at rates required for autophagosome biogenesis in vivo.

To address these questions and explore the underlying PLT mechanisms at PERCS, we first asked whether membrane assembly is rate-limiting for autophagosome biogenesis. We performed time-lapse fluorescence imaging of living yeast cells expressing 2GFP-Atg8, which is covalently attached to autophagic membranes, after 1 h of nitrogen starvation (hereafter starvation; Huang et al., 2000; Kirisako et al., 1999). For each biogenesis event, we measured the size of the forming autophagosome at the point of maximal diameter (AP size; Fig. 1, D–F). Within the observed autophagosome sizes, changes in diameter are linearly correlated with increasing surface area or the number of phospholipids of autophagosomes (linear regression, $r = 0.995$; Fig. 1 B). In parallel to size, we determined the time from detecting an Atg8 punctum (start), which represents the formation of an Atg8ylated small phagophore, until the time point of maximal size of a ring-shaped structure (duration$_{max}$), which includes the majority of the phagophore expansion phase of cup-shaped phagophores during imaging. In addition, we measured the time until the ring-shaped autophagosome disappeared upon vacuolar fusion (end) to capture the duration of the whole autophagosome biogenesis event (duration$_{total}$; Fig. 1, D–F). Both the average duration$_{total}$ and size of forming autophagosomes were consistent with previous observations (Fig. S1, A and B; Schütter et al., 2020; Xie et al., 2008). Interestingly, our analysis revealed only a weak positive correlation between the duration$_{max}$ or duration$_{total}$ and size of forming autophagosomes (Fig. 1, E and F). We observed a broad range of durations for autophagosomes of similar size (Fig. 1, E and F), indicating the resulting size of a forming autophagosome does not strictly determine the duration of its expansion phase or complete biogenesis and vice versa. These data indicate that autophagic membrane

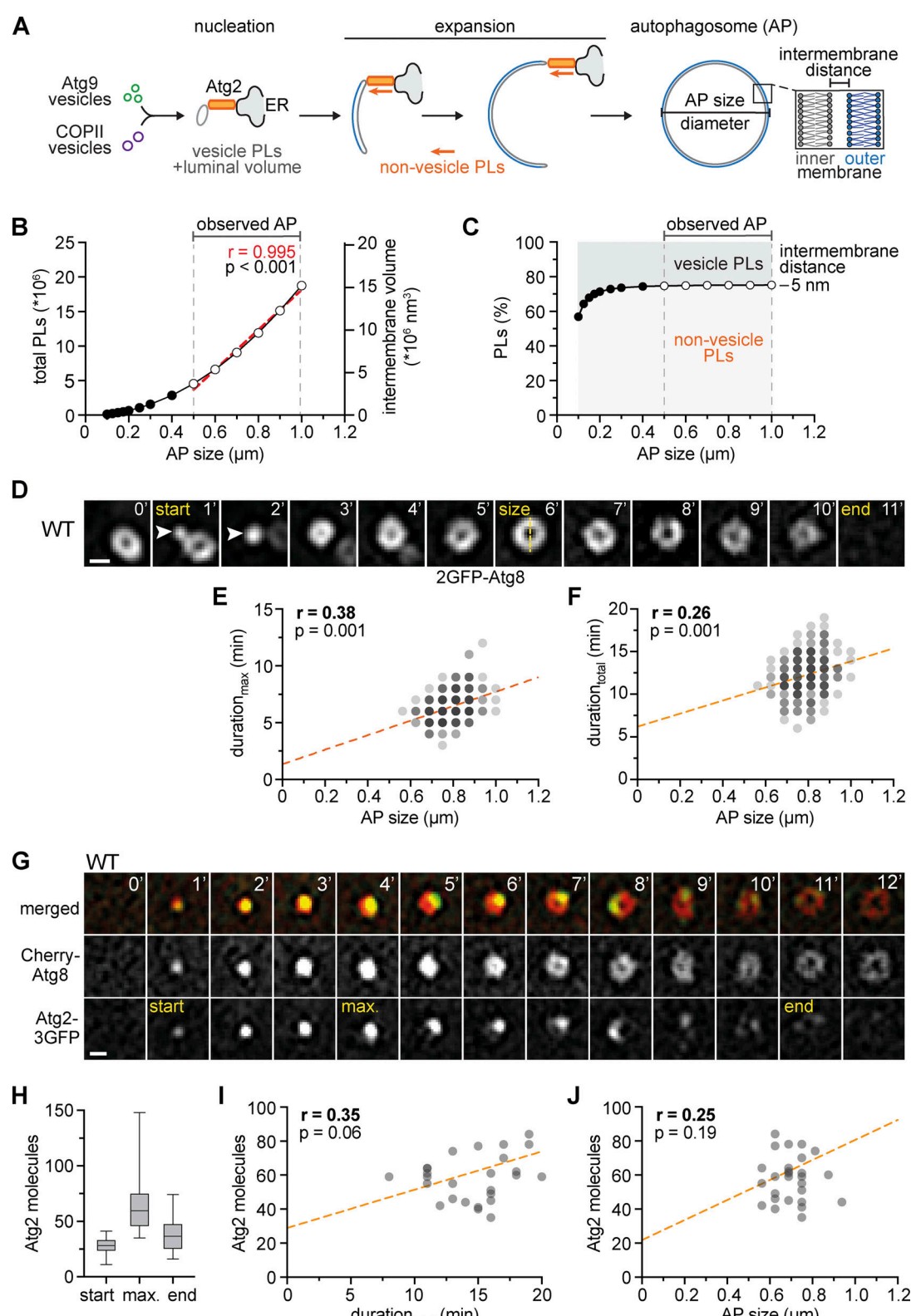

Figure 1. **Membrane assembly and Atg2 at PERCS are not rate-limiting for autophagosome formation. (A)** Model for autophagosome biogenesis based on vesicle- and non-vesicle PLT mechanisms. **(B)** Number of phospholipids and intermembrane volume needed to form autophagosomes of indicated diameter (AP size) is based on an intermembrane distance of 5 nm (Bieber et al., 2022) and calculated as described in Materials and methods. The red regression line and P value indicate a near linear relationship between diameter (AP size) and both parameters. **(C)** The relative quantitative contribution of vesicle- and non-vesicle phospholipid transfer to autophagosomes. Based on the total phospholipid content of three Atg9 vesicles (60 nm) and the corresponding number of COPII vesicles (80 nm) required to form the intermembrane volume of autophagosomes of a given size, we calculated the number of phospholipids derived from non-vesicle phospholipids as described in Materials and methods. **(D)** Timelapse fluorescence imaging of WT cells expressing *2GFP-ATG8* after starvation

(1 h). Representative timeline of autophagosome biogenesis is shown as single Z-sections. Arrowheads indicate nucleated phagophores. Scale bar is 0.5 μm. **(E and F)** Quantification of data is shown in D. **(E)** Simple linear regression of duration$_{max}$ (time from nucleation until maximum size) and autophagosome (AP) size (number of independent experiments, $n$ = 6; 230 events analyzed in total). **(F)** Simple linear regression of duration$_{total}$ (time from nucleation until disappearance) and autophagosome (AP) size (number of independent experiments, $n$ = 6; 230 events analyzed in total). **(G)** Timelapse fluorescence imaging of WT cells expressing *Cherry-ATG8* and *ATG2-3GFP* after starvation (1 h). Images are single Z-sections. Representative timeline of autophagosome biogenesis is shown. Atg2 signal intensity was measured at the indicated time points: start, maximum ("max"), and end. Scale bar is 0.5 μm. **(H–J)** Quantification of data shown in G (number of independent experiments, $n$ = 3; 30 events analyzed in total). **(H)** Number of Atg2 molecules at indicated timepoints. Simple linear regression of maximum number of Atg2 molecules and duration$_{total}$ (I) or autophagosome (AP) size (J).

---

assembly is not a generally rate-limiting factor for autophagosome formation.

According to our quantitative model, non-vesicular PLT contributes the majority of phospholipids to autophagosome biogenesis (Fig. 1, A–C). We hypothesized that membrane assembly and, as a consequence, duration and/or size of forming autophagosomes may be determined by the number of Atg2 molecules engaged in phospholipid channeling at PERCS. To test our predictions, we first asked whether the PLT activity of Atg2 is required for autophagosome biogenesis. We genomically integrated either a wild-type Atg2 or full-length PLT-deficient Atg2$^{ΔPLT}$ variant C-terminally tagged with mCherry into the genomic *ATG2* locus. The Atg2$^{ΔPLT}$ variant carries 12 amino acid exchanges in its N-terminal region modeled after PLT-deficient mini variants of human ATG2A (Fig. S1 C; Valverde et al., 2019). As we observed lower steady-state protein levels for Atg2$^{ΔPLT}$ compared with Atg2 (Fig. S1 D), we generated an Atg2 variant expressed under the control of the *ATG23* promoter (Atg2$^{low}$-2Cherry) with slightly lower protein steady-state levels than Atg2$^{ΔPLT}$ (Fig. S1 D). Strikingly, cells expressing Atg2 or Atg2$^{low}$ showed a similar number of Atg8 puncta, number and size of autophagosomes, and autophagy flux, but Atg2$^{ΔPLT}$ failed to support the formation of any detectable autophagosomes or autophagy flux (Fig. S1, E and F). These data suggest an essential role for Atg2-mediated PLT during autophagosome biogenesis in vivo, an evolutionarily conserved feature (Tan and Finkel, 2022; Valverde et al., 2019).

Second, we tested whether Atg2-mediated PLT is rate limiting for autophagosome biogenesis. We analyzed cells coexpressing genomically N-terminally mCherry-tagged Atg8 (Cherry-Atg8) and C-terminally triple GFP-tagged Atg2 (Atg2-3GFP) and measured fluorescence intensities of Atg2 in addition to the duration$_{total}$ and size of forming autophagosomes for each biogenesis event after 1 h of starvation (Fig. 1 G). To determine the absolute number of Atg2 molecules, we normalized fluorescence signals of Atg2-3GFP associated with Atg8-marked autophagic membranes to the puncta intensities of Cse4-GFP expressing cells (Fig. S1 G). A single kinetochore cluster contains ∼80 copies of Cse4-GFP, which can be used to normalize and convert GFP fluorescence signals into a number of proteins (Fig. S1 G; Yamamoto et al., 2012). Following the number of Atg2 molecules at Atg8-marked structures over time, we detected a strikingly dynamic behavior of Atg2 during autophagosome biogenesis (Fig. 1, G and H; and Fig. S1 H), consistent with previous observations for mammalian ATG2A (Sakai et al., 2020). Around 28 ± 7 molecules of Atg2 coemerged with Atg8-positive punctate phagophores ("start"; Fig. 1, G and H), suggesting Atg8ylation and Atg2 recruitment closely coincide in a temporal manner.

Interestingly, previous work estimated roughly that three Atg9 vesicles carrying in total ∼27 trimeric Atg9 complexes are incorporated into the early phagophore (Maeda et al., 2020; Yamamoto et al., 2012). Given that Atg2 and Atg9 physically interact (Gómez-Sánchez et al., 2018; van Vliet et al., 2022), the number of Atg2 molecules may be physically coupled with the number of trimeric Atg9 complexes in a 1:1 ratio at early phagophores to coordinate PLT with phospholipid scrambling. Following this initial stage, the number of Atg2 molecules increased to a maximum of 64 ± 23 Atg2 molecules during phagophore expansion ("max") and then dropped to 38 ± 15 Atg2 molecules at closing or closed autophagosomes ("end"; Fig. 1, G and H). Importantly, despite the observed increase of Atg2 molecules during phagophore expansion, we did not detect any significant correlation between the maximum number of Atg2 molecules and either the duration$_{total}$ or the size of the corresponding forming autophagosomes (Fig. 1, I and J). These data strongly support the conclusion that, although essential, Atg2-mediated PLT is not a rate-limiting factor for the dynamics of autophagosome formation.

Our observations raise the possibility that Atg2 clusters at PERCS may provide sufficiently high in vivo PLT capacity to drive autophagosome biogenesis in a non-rate-limiting manner. Alternatively, but not mutually exclusively, additional PLT proteins may contribute to non-rate-limiting PLT into expanding phagophores in parallel to Atg2. To test whether additional PLT proteins drive autophagy, we turned to the conserved membrane tether and PLT protein Vps13, which shares key structural features with Atg2 and functions at various organelle contact sites in yeast and mammalian cells (Fig. 2 A; Bean et al., 2018; Kumar et al., 2018; Li et al., 2020). Additionally, Vps13 proteins have been previously linked to autophagy. VPS13A has been implicated in mammalian autophagy, Vps13D plays a role in autophagy during intestinal development in *Drosophila*, and yeast Vps13 has been linked to selective turnover of ER and mitochondria by autophagy by unknown mechanisms (Anding et al., 2018; Chen et al., 2020; Muñoz-Braceras et al., 2015; Park et al., 2016; Shen et al., 2021; Yeshaw et al., 2019).

Using fluorescence imaging, we examined whether Vps13 is spatially associated with autophagic membranes. Consistent with previous data, we could visualize Vps13 in cells upon roughly fourfold overexpression of plasmid-encoded *VPS13* internally tagged with GFP (Vps13^GFP; Fig. 2 B and Fig. 5 B; Bean et al., 2018). Importantly, similar to Atg2, we detected Vps13^GFP at around 80% of autophagic structures marked with 2Cherry-Atg8 after 1 h of starvation (Fig. 2 B). Using a non-overexpressed Vps13^2GFP variant integrated into the endogenous *VPS13* locus, we were able to capture a few events, which

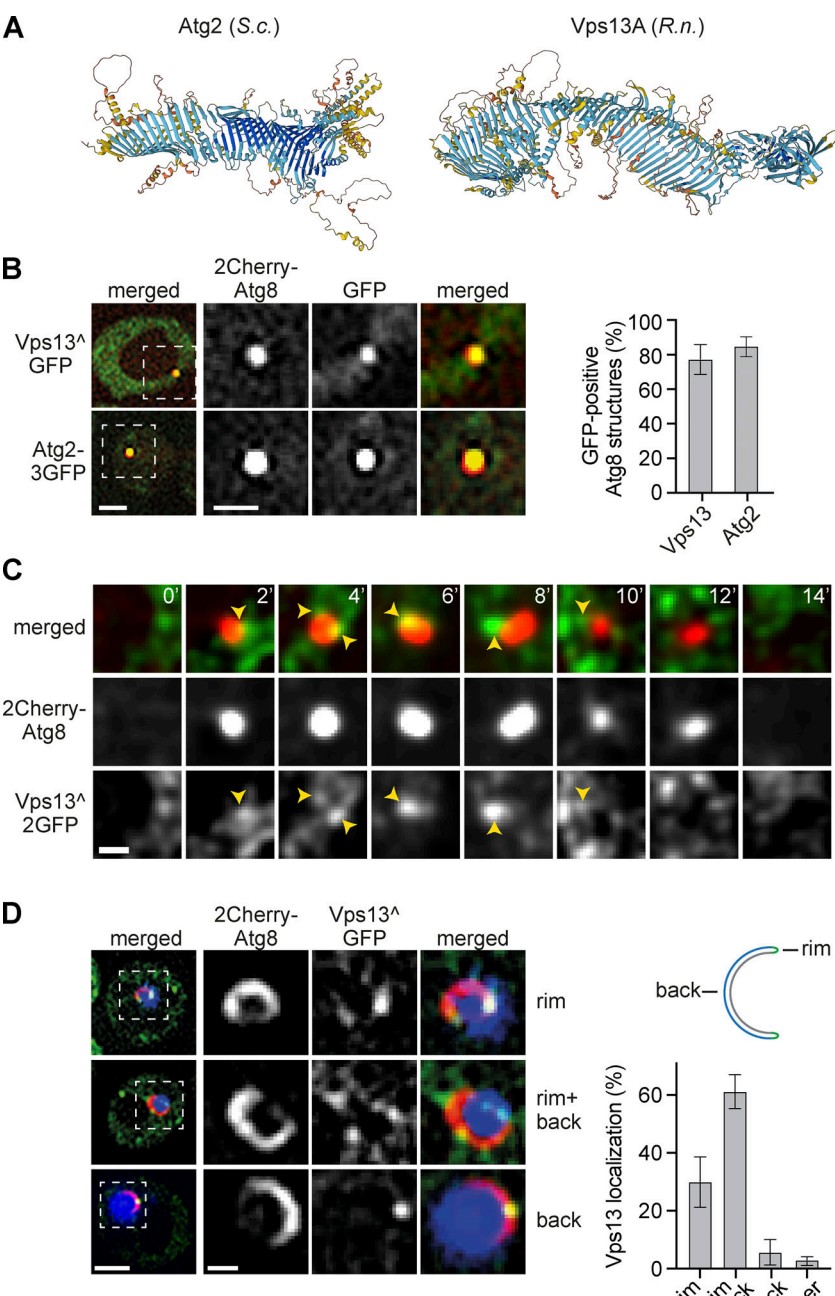

**Figure 2. Vps13 quantitatively localizes to the phagophore rim. (A)** AlphaFold-based structure predictions of *S. cerevisiae* Atg2 and *Rattus norvegicus* Vps13A (Jumper et al., 2021). **(B)** Fluorescence imaging of Δ*vps13* cells expressing *2Cherry-ATG8* and pRS423-*VPS13^GFP* or cells expressing *Cherry-ATG8* and *ATG2-3GFP* after starvation (1 h). The analysis of Vps13 and Atg2 was performed in independent experiments. Quantified data are shown in the right panel (number of independent experiments, *n* = 4; 200 structures analyzed in total). Scale bars are 2 and 1 μm (zoom in). **(C)** Timelapse fluorescence imaging of WT cells expressing *VPS13^2GFP* and *2Cherry-ATG8* after starvation (1 h). Representative timeline for autophagosome biogenesis is shown as a single section. Scale bar is 0.5 μm. **(D)** Fluorescence imaging of Δ*vps13* cells expressing *2Cherry-ATG8*, pRS423-*VPS13^GFP*, and pRS425-*APE1-BFP* after starvation (1 h). Quantified data are shown in the right panel (number of independent experiments, *n* = 4; 200 structures analyzed in total). Scale bars are 2 and 1 μm (zoom in).

showed a continuous association of Vps13 with forming autophagosomes during fluorescence timelapse analysis (Fig. 2 C). These data indicate quantitative spatial and temporal association of Vps13 with forming autophagosomes. Atg2 specifically localizes to the rim of expanding phagophores (Gómez-Sánchez et al., 2018; Graef et al., 2013; Sakai et al., 2020; Suzuki et al., 2013). To examine the localization of Vps13 at expanding phagophores, we overexpressed tagBFP-prApe1 in cells coexpressing 2Cherry-Atg8 and overexpressed Vps13^GFP. Overexpressed tagBFP-prApe1 oligomerizes into large cytosolic clusters forcing the formation of enlarged phagophores (Pfaffenwimmer et al., 2014; Suzuki et al., 2013). Strikingly, Vps13 localized to the rim of ~90% of enlarged phagophores (Fig. 2 D). We detected Vps13 at the rim alone or at the rim and the convex backside of

phagophores in ~30 or ~60% of cases, respectively (Fig. 2 D). These data place Vps13 at the rim of expanding phagophores in parallel with Atg2. In contrast to Atg2, Vps13 also localizes to the phagophore backside likely at phagophore–vacuole contacts (Bieber et al., 2022; Graef et al., 2013; Suzuki et al., 2013).

Our data ideally positions Vps13 spatially to promote phagophore expansion in parallel with Atg2. To test whether Vps13 functions in autophagosome biogenesis, we measured autophagy in WT and Δ*vps13* cells expressing 2GFP-Atg8 during starvation. Interestingly, we observed a slightly increased number of Atg8 puncta and autophagosomes in Δ*vps13* cells compared with WT cells without effects on autophagosome size or resulting autophagy flux (Fig. S2, A–D), consistent with previous work (Chen et al., 2020). These data demonstrate that, in

contrast to Atg2, Vps13 is not essential for autophagosome formation. However, the combination of an elevated number of autophagosomes without a proportionate increase of autophagy flux suggested that autophagosomes form more slowly in the absence of Vps13.

To test for the role of Vps13 in autophagosome biogenesis dynamics, we performed time-lapse fluorescence imaging in WT and Δvps13 cells expressing 2GFP-Atg8. Consistent with our hypothesis, we detected a moderate but significant increase in the duration$_{max}$ or duration$_{total}$ of autophagosome biogenesis in the absence of Vps13 (Fig. 3, A, B, and E). Importantly, we observed a strongly increased positive correlation between the duration$_{max}$ or duration$_{total}$ and size of forming autophagosomes in Δvps13 cells compared with WT cells (Fig. 3, C, D, F, and G), suggesting rate-limiting membrane assembly during autophagosome biogenesis in the absence of Vps13. To directly assess the effects on phagophore expansion, we determined the correlation between duration and final size of phagophores forming on enlarged tagBFP-preApe1 oligomers in 2GFP-Atg8 expressing WT and Δvps13 cells (Fig. 3 H). We found a significant positive correlation for Δvps13 cells in contrast to WT cells (Fig. 3, I and J). Taken together, these findings indicate that Vps13 functions in parallel to Atg2 during phagophore expansion and that membrane assembly becomes a limiting factor for autophagosome formation in the absence of Vps13.

Three known adaptor proteins, Mcp1, Ypt35, and Spo71, recruit Vps13 to mitochondria, endosome, and meiotic prospore membranes, respectively, in a dynamic and competitive manner (Bean et al., 2018). To test whether these adaptors play a role in Vps13 recruitment to phagophores, we analyzed the localization of overexpressed Vps13^GFP to autophagic structures marked by 2Cherry-Atg8 in the presence or absence of the Vps13 adaptor proteins. Vps13^GFP spatially associated with autophagic structures in an indistinguishable manner in WT or Δmcp1Δypt35Δspo71 (ΔΔΔ) cells after 1 h of starvation (Fig. S2 E). The absence of Ypt35 alone or of all three adaptors did not affect the localization of Vps13 at expanding phagophores (Fig. S2 H). Taken together, these data demonstrate that known adaptors are not required for Vps13 recruitment to autophagic membranes nor define the position of Vps13 at forming autophagosomes. To begin to examine how Vps13 is recruited to autophagic membranes, we generated a Vps13 variant (Vps13$^{ΔC}$^GFP) lacking the last 647 aa at the C-terminus previously implicated in membrane binding (De et al., 2017; Kolakowski et al., 2020; Rzepnikowska et al., 2017). Interestingly, we observed a significant decrease in Vps13 association with autophagic membranes in the absence of the C-terminal domains (Fig. S2, F and G), indicating a critical function for Vps13 recruitment to autophagic membranes.

To challenge our model, we examined potentially indirect effects on autophagosome biogenesis caused by the absence of Vps13 from known organelle contact sites. WT and ΔΔΔ cells expressing 2GFP-Atg8 showed the same autophagy flux during starvation (Fig. S2 I). Importantly, in contrast to Δvps13 cells, timelapse fluorescence imaging showed a reduced correlation of duration$_{total}$ and size of forming autophagosomes in ΔΔΔ cells compared with WT cells (Fig. S2 J). These data indicate the absence of Vps13 directly affects the dynamics of autophagosome

biogenesis independently of impaired functions of Vps13 at other known organelle contact sites.

Based on our data, we hypothesized that Atg2-mediated PLT at PERCS limits membrane assembly during autophagosome biogenesis in the absence Vps13. We probed whether the number of Atg2 molecules correlated with the size of forming autophagosomes in WT and Δvps13 cells expressing Cherry-Atg8 and Atg2-3GFP by time-lapse fluorescence imaging after 1 h of starvation. Atg2 localized to phagophores in a Vps13-independent manner (Fig. 4, A and B). Importantly, in contrast to WT cells, we determined a strong positive correlation between the number of Atg2 molecules and autophagosome size in Δvps13 cells (Fig. 4, C and D). These data suggest that the total PLT capacity of Atg2 clusters at PERCS becomes limiting in the absence of Vps13. Notably, the linear regressions in our correlation analyses of duration or Atg2 molecules and size of autophagosomes cut respective x-axes at a value of around 100 nm in the absence of Vps13 (Fig. 3 G and Fig. 4 D), consistent with phagophore nucleation from Atg9- and COPII-vesicles and subsequent phagophore expansion by Atg2-dependent non-vesicular PLT.

The in vivo PLT rates of Atg2 molecules at PERCS are unknown. Significant positive correlations between the duration and size of forming autophagosomes and the number of Atg2 molecules in the absence of Vps13 allowed us to determine an apparent in vivo PLT rate for Atg2. We derived the number of phospholipids transported by Atg2 from the size of autophagosomes and the theoretical contribution of non-vesicle PLT of 75% in relation to the maximum number of Atg2 molecules at phagophores and the duration for each biogenesis event as described in materials and methods (Fig. 1, B and C). Interestingly, we determined an average in vivo PLT rate of ~200 phospholipids per Atg2 molecule and second (Fig. 4 E). This number is in remarkable agreement with theoretical estimates for in vitro PLT rates of 100–750 phospholipids per molecule and second of purified yeast and mammalian Atg2 (von Bülow and Hummer, 2020 Preprint). Taken together, these data suggest that the PLT capacity of Atg2 clusters at PERCS alone may be sufficient but becomes limiting for autophagosome biogenesis in the absence of Vp13.

Our data suggest that Vps13 accelerates autophagic membrane assembly to a non-rate-limiting level. To define whether Vps13 promotes autophagosome biogenesis by virtue of its PLT activity, we analyzed either a wild-type or a previously described VPS13 variant with defective PLT activity (vps13$^{ΔPLT}$, termed M2 in Li et al. [2020]). Similar to Vps13^GFP, the overexpressed PLT-deficient Vps13$^{ΔPLT}$ variant carrying an internal GFP tag (Vps13$^{ΔPLT}$^GFP) quantitatively associated with autophagic structures (Fig. 5, A and B). A genomically integrated vps13$^{ΔPLT}$ variant carrying an internal HA tag (Vps13$^{ΔPLT}$^HA) did not affect the number of Atg8 puncta, number, duration$_{total}$ or size of autophagosomes, or autophagy flux (Fig. S3, A–E). Importantly, when we analyzed the dynamics of autophagosome biogenesis in VPS13^HA or vps13$^{ΔPLT}$^HA cells expressing 2GFP-Atg8, we observed a significantly increased positive correlation between the duration$_{max}$ or duration$_{total}$ and size of forming autophagosomes in the absence of Vps13-mediated PLT (Fig. 5, C–G). These data strongly support the conclusion that the PLT by Vps13 drives autophagic membrane assembly in parallel to Atg2.

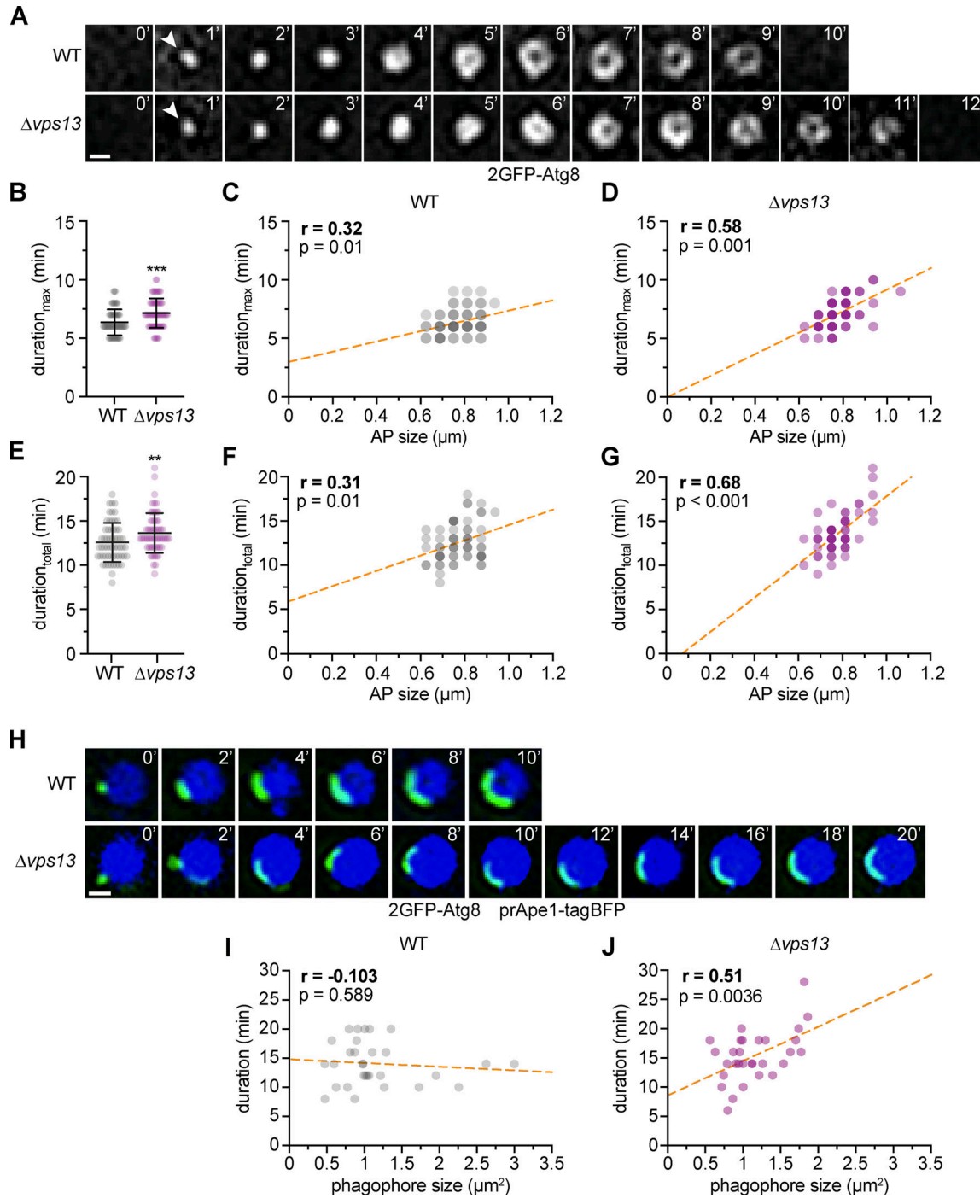

Figure 3. **Rate-limiting membrane assembly in the absence of Vps13. (A)** Time-lapse fluorescence imaging of WT and Δ*vps13* cells expressing *2GFP-ATG8* after starvation (1 h). Representative timelines for autophagosome biogenesis events are shown as single Z-sections. WT data were included in Fig. 1, D–F (number of independent experiments, *n* = 6; 60 events/strain analyzed in total). Scale bar is 0.5 μm. **(B)** Duration$_{max}$ of autophagosome biogenesis for data shown in A. **(C and D)** Simple linear regressions of duration$_{max}$ and autophagosome size for WT (C) and Δ*vps13* (D) cells. **(E)** Duration$_{total}$ of autophagosome biogenesis for data shown in A. **(F and G)** Simple linear regressions of duration$_{total}$ and autophagosome size for WT (F) and Δ*vps13* (G) cells. **(H)** Timelapse fluorescence imaging of WT and Δ*vps13* cells expressing *2GFP-ATG8* and *pRS425-APE1-BFP* after starvation (1 h). Representative timelines for phagophore expansion events are shown (number of independent experiments, *n* = 3; 30 events/strain analyzed in total). Scale bar is 1 μm. **(I and J)** Simple linear regressions of duration and phagophore size for WT (I) and Δ*vps13* (J) cells.

In summary, our work indicates that after nucleation from Atg9 and COPII vesicles, phagophores expand in a generally non-rate-limiting manner during autophagosome biogenesis. Importantly, cells depend on the parallel activity of the conserved tether and PLT proteins Atg2 and Vps13 for non-rate-limiting PLT across phagophore–ER contacts (Fig. 5 H).

Three adaptors compete to recruit a limiting number of Vps13 molecules to various organelle contacts (Bean et al., 2018; Lang

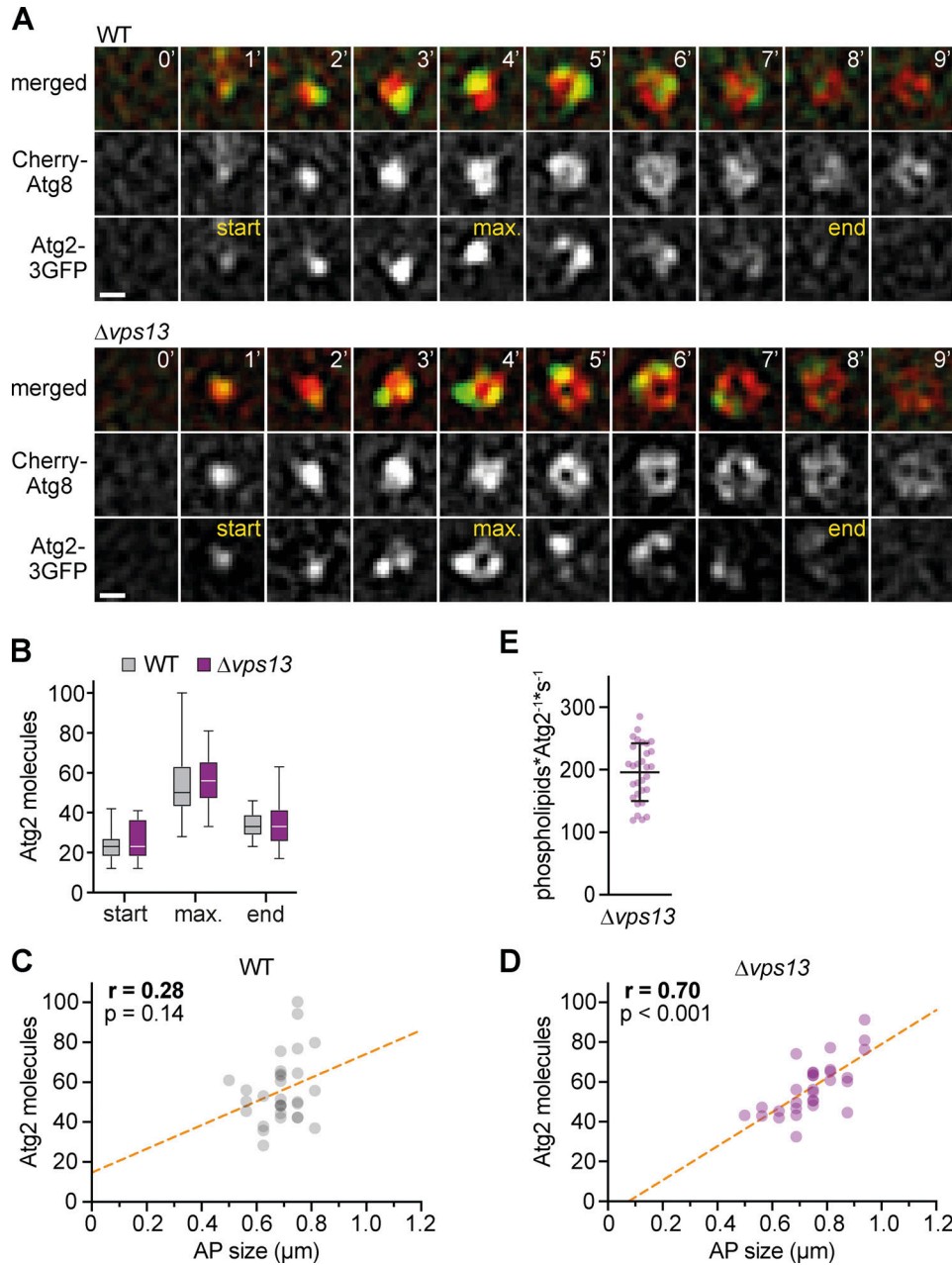

Figure 4. **Rate-limiting number of Atg2 proteins during autophagosome biogenesis in the absence of Vps13. (A)** Time-lapse fluorescence imaging of WT and Δ*vps13* cells expressing *Cherry-ATG8* and *ATG2-3GFP* after starvation (1 h; *n* = 3; 30 events/strain) shown as single Z-sections. Scale bar is 0.5 μm. **(B–D)** Quantification of data is shown in A. **(B)** Number of Atg2 molecules at indicated timepoints: start, maximum (max), and end. **(C and D)** Simple linear regression of the number of Atg2 molecules and autophagosome (AP) size for WT (C) and Δ*vps13* cells (D). **(E)** Number of phospholipids transferred per Atg2 molecule and second in vivo. Calculations are based on the model shown in Fig. 1, A–C, and the duration and size of autophagosomes and the number of Atg2 molecules for each event shown in A–D.

et al., 2015; Park et al., 2016). Thus, cells may tune phospholipid flux into different organelles by differential enrichment of Vps13 at corresponding organelle contact sites. Based on such a scenario, phagophore-bound Vps13 may function to preferentially direct phospholipids into autophagosome biogenesis in addition to other mechanisms including localized phospholipid synthesis (Schütter et al., 2020). Thus, the parallel function of members of the bridge-like PLT protein family at the same organelle contact sites may emerge as a general

mechanism for how cells control lipid fluxes between organelles in quantity and quality.

We provide a first estimate for an apparent in vivo PLT rate of Atg2 during autophagosome biogenesis. However, it remains to be analyzed whether all Atg2 molecules associated with phagophores form contacts with phagophore or ER membranes and engage in PLT across PERCS. Furthermore, we do not exclude the possibility that additional PLT proteins other than Atg2 and Vps13 and/or continuous membrane contacts may

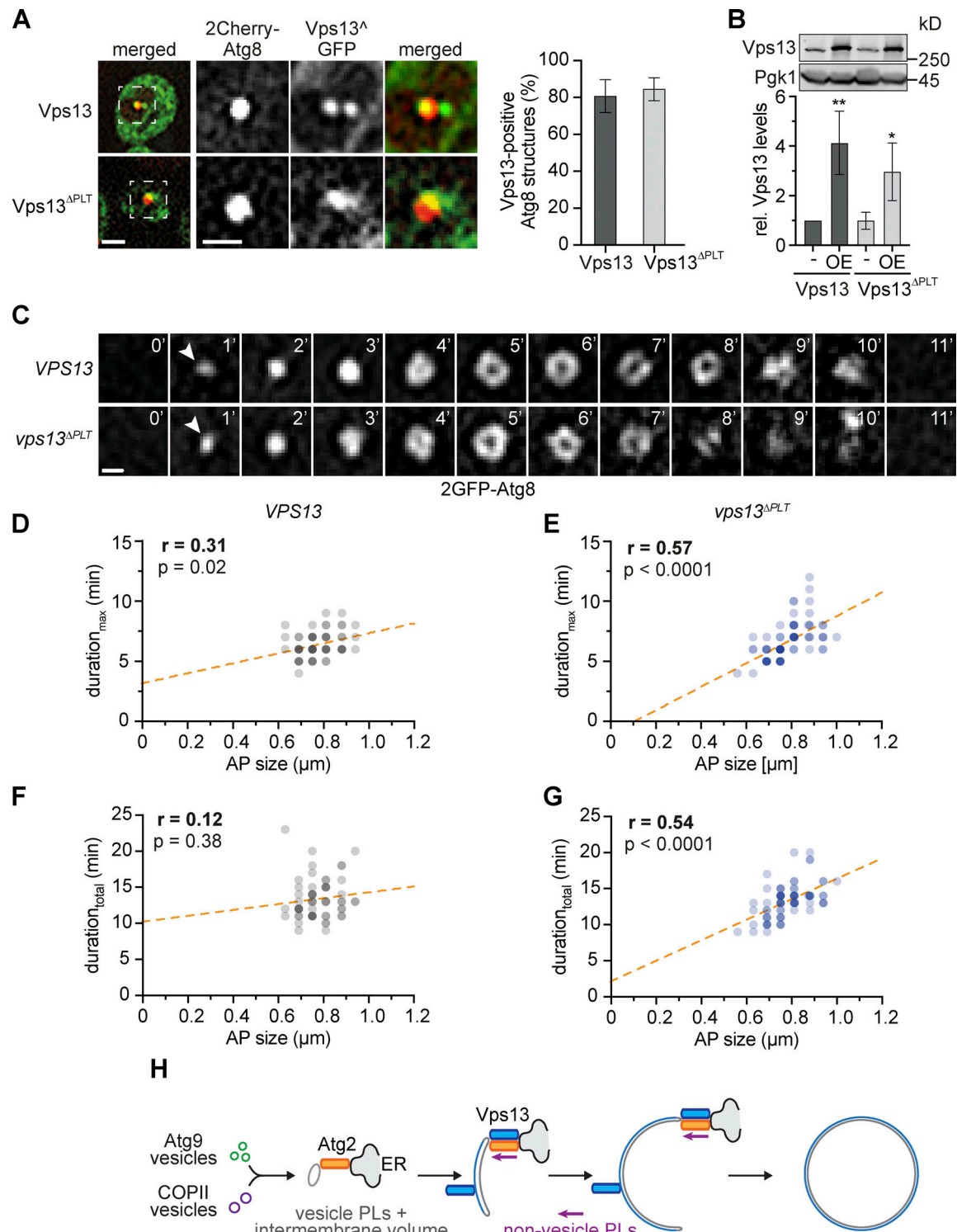

Figure 5. **Vps13-mediated PLT is required for non-rate-limiting membrane assembly during autophagosome biogenesis. (A)** Fluorescence imaging of indicated strains expressing *2Cherry-ATG8* and pRS423-*VPS13^GFP* in Δ*vps13* background after starvation (1 h). Right panel shows quantified data (number of independent experiments, *n* = 4; 200 structures/strain analyzed in total). Scale bars are 2 and 1 µm (zoom in). **(B)** Normalized protein levels of genomic and plasmid-based overexpressed (OE, pRS423) Vps13^GFP and Vps13$^{\Delta PLT}$^GFP in growing cells analyzed by whole-cell extraction and Western blot quantification using α-Vps13 and α-Pgk1 antibodies. **(C)** Time-lapse fluorescence imaging of WT and *vps13*$^{\Delta PLT}$ cells expressing *2GFP-ATG8* after starvation (1 h; number of independent experiments, *n* = 3; 30 events/strain analyzed in total). Representative timelines of autophagosome biogenesis are shown as single Z-sections. Scale bar is 0.5 µm. WT data were included in Fig. 1, D–F. **(D and E)** Quantification of data shown in C. **(D and E)** Simple linear regression of duration$_{max}$ and autophagosome (AP) size in WT (D) and *vps13*$^{\Delta PLT}$ cells (E). **(F and G)** Simple linear regression of duration$_{total}$ and autophagosome (AP) size in WT (F) and *vps13*$^{\Delta PLT}$ cells (G). **(H)** Model for parallel PLT via the conserved PLT proteins Vps13 and Atg2 driving phagophore expansion after nucleation from Atg9 and COPII vesicles during autophagosome biogenesis. Source data are available for this figure: SourceData F5.

contribute to non-vesicular PLT into forming autophagosomes (Hayashi-Nishino et al., 2009; Neuman et al., 2022; Ylä-Anttila et al., 2009). However, a remarkable similarity between our estimates for the in vivo PLT rate of Atg2 and the in vitro data suggests that, in principle, Atg2 assemblies at PERCS alone may possess sufficient albeit limiting PLT capacity for promoting autophagosome biogenesis (von Bülow and Hummer, 2020 Preprint). Interestingly, while parallel PLT drives the dynamics of autophagosome biogenesis, the function of Atg2 and Vps13 for autophagosome biogenesis is not equivalent. Only Atg2-mediated PLT is essential for autophagy in yeast and mammalian cells (Tan and Finkel, 2022; Valverde et al., 2019). The molecular basis for the essential nature of PLT by Atg2 is unclear, but the physical Atg2–Atg9 interaction raises the possibility that phospholipid transfer by Atg2 has to be coupled to the scramblase activity of Atg9 (Gómez-Sánchez et al., 2018; Maeda et al., 2020; Matoba et al., 2020; Orii et al., 2021; van Vliet et al., 2022). It remains to be analyzed whether Vps13 binds to a dedicated scramblase or relies directly or indirectly on Atg9-mediated scrambling within the phagophore membrane. Recent work has identified that yeast Vps13 and human VPS13A are coupled with the scramblases Mcp1 in mitochondria and XK in the plasma membrane, respectively (Adlakha et al., 2022; Guillén-Samander et al., 2022; Park et al., 2022). Human VPS13A-D variants are associated with distinct pathologies (Ugur et al., 2020). Our work now opens up the possibility that VPS13 variants directly affect autophagosome biogenesis, which may contribute to specific pathologies.

## Material and methods

### Strains and media

All *Saccharomyces cerevisiae* strains used in this study are derivatives of W303 and are listed in Table 1. To generate gene deletions, full open reading frames (ORFs) were replaced by respective marker cassettes using PCR-based targeted homologous recombination (Longtine et al., 1998). Genes were modified to express functional C-terminally tagged variants via PCR-based targeted homologous recombination using pFA6a-link-yEGFP-CaURA3, pFA6a-link-3yEGFP-CaURA3, pFA6a-link-mCherry-kanMX6, and pFA6a-link-mCherry-His3MX6 as described previously (Graef et al., 2013; Sheff and Thorn, 2004). Plasmids used in this study are listed in Table 2.

For the generation of the Δvps13::VPS13^HA-natMX6, Δvps13::VPS13^2GFP-CaUra3, and Δvps13::vps13^ΔPLT^HA-natMX6 strains, a 422 bp 5′-upstream fragment of the genomic VPS13 locus, full-length VPS13, VPS13^2GFP containing two internal GFP inserted between residue 499 and 500 or vps13^ΔPLT (M2 in Li et al. [2020]) containing internal triple HA tag inserted after residue 499 (Lang et al., 2015), and a 501 bp 3′-downstream fragment of the genomic VPS13 locus were assembled in a pRS315 plasmid backbone using gap repair cloning. Isolated and sequenced pRS315-VPS13^HA-natMX6, pRS315-VPS13^2GFP-CaUra3, and pRS315-vps13^ΔPLT^HA-natMX6 plasmids were cut with XhoI and NotI and transformed into pRS306-2GFP-ATG8 Δvps13::His3MX6 cells. Homologous recombination replaced the Δvps13::His3MX6 cassette with the VPS13^HA-natMX6, VPS13^2GFP-CaUra3, or

vps13^ΔPLT^HA-natMX6 constructs at the endogenous VPS13 locus, respectively, giving rise to Δvps13::VPS13^HA-natMX6, Δvps13::VPS13^2GFP-CaUra3, and Δvps13::vps13^ΔPLT^HA-natMX6.

To generate the Δatg2::atg2^ΔPLT-mCherry-natMX6 strain, we assembled a 1,000 bp 5′-upstream fragment of the genomic ATG2 locus, the atg2^ΔPLT gene variant, which contains amino acid

Table 1.  **S. cerevisiae strains used in this study**

| S. cerevisiae | Genotype | Ref. |
|---|---|---|
| YMG2355 | W303 pRS306-2GFP-ATG8-CaURA3 Δvps13::His3MX6 | This manuscript |
| YMG2358 | W303 pRS306-2GFP-ATG8 kanMX6-pr^ATG23-ATG2-2xmCherry-His3MX6 | This manuscript |
| YMG2359 | W303 pRS306-2GFP-ATG8 ATG2-mCherry-His3MX6 | This manuscript |
| YMG2364 | W303 ATG2-3GFP-CaURA3 pRS304-mCherry-ATG8 Δatg19::His3MX6 Δvps13::natMX6 | This manuscript |
| YMG2393 | W303 pRS306-2GFP-ATG8 Δvps13::VPS13^HA-natMX6 | This manuscript |
| YMG2395 | W303 pRS306-2GFP-ATG8 Δvps13::vps13^HA-M2-natMX6 (vps13^ΔPLT) | This manuscript |
| YMG2401 | W303 pRS306-2GFP-ATG8 Δmcp1::kanMX6 Δypt35::natMX6 Δspo71::TRP1 | This manuscript |
| YMG2409 | W303 CSE4-GFP-CaURA3 | This manuscript |
| YMG2419 | W303 pRS306-2GFP-ATG8 Δatg2::atg2^ΔPLT-mCherry-kanMX6 | This manuscript |
| YMG2422 | W303 Δatg8::2mCherry-ATG8-hphMX6 Δvps13::TRP1 pRS423-VPS13^GFP | This manuscript |
| YMG2424 | W303 Δatg8::2mCherry-ATG8-hphMX6 Δvps13::TRP1 pRS423-VPS13^GFP-M2 (vps13^ΔPLT^GFP) | This manuscript |
| YMG2425 | W303 Δatg8::2mCherry-ATG8-hphMX6 Δvps13::TRP1 Δmcp1::kanMX6 Δypt35::natMX6 Δspo71::CaURA3 pRS423-VPS13^GFP | This manuscript |
| YMG2426 | W303 Δatg8::2mCherry-ATG8-hphMX6 Δvps13::natMX6 pRS423-VPS13^GFP pRS425-prCUP1-APE1-prApe1-tagBFP-APE1 | This manuscript |
| YMG2857 | W303 pRS306-2GFP-ATG8 | Velázquez et al., 2016 |
| YMG2858 | W303 ATG2-3GFP-CaUAR3 pRS304-mCherry-ATG8 Δatg19::His3MX6 | This manuscript |
| YRD391 | W303 Δatg8::2mCherry-ATG8-hphMX6 Δvps13::VPS13^2GFP-CaUra3 | This manuscript |
| YRD402 | W303 Δatg8::2mCherry-ATG8-hphMX6 Δvps13::TRP1 + pRS423-VPS13 ^ΔC^GFP | This manuscript |

Table 2.  **Plasmids used in this study**

| E. coli | Plasmid | Ref. |
|---|---|---|
| EMG835 | pRS425-prCUP1-APE1-prAPE1-tagBFP-APE1 | Schütter et al., 2020 |
| EMG544 | pRS423-VPS13^GFP | Li et al., 2020 |
| EMG548 | pRS423-VPS13^GFP-M2 (vps13^ΔPLT^GFP) | Li et al., 2020 |
| EMG581 | pRS423-VPS13^ΔC^GFP | This manuscript |

exchanges L18E (CTT > gaa), L69E (CTT > gaa), F88D (TTT > gat), L176K (TTG > aag), L180R (TTA > aga), V201E (GTT > gaa), I218K (ATA > aag), I220K (ATT > aag), L298E (TTG > gag), F302E (TTT > gaa), V323R (GTT > aga), I345D (ATT > gat), a C-terminal mCherry-kanMX6 fragment derived from pFA6a-link-mCherry-kanMX6, and a 1,000 bp 3′-downstream fragment of the genomic ATG2 locus in a pRS315 plasmid backbone using gap repair cloning. The isolated and sequenced pRS315-atg2$^{\Delta PLT}$-mCherry-kanMX6 plasmid was cut with NotI and SpeI and transformed into pRS306-2GFP-ATG8 Δatg2::natMX6 cells. Homologous recombination replaced the Δatg2::natMX6 cassette with the atg2$^{\Delta PLT}$-mCherry-kanMX6 construct giving rise to Δatg2::atg2$^{\Delta PLT}$-mCherry-natMX6 at the endogenous ATG2 locus.

To generate pRS423-VPS13$^{\Delta C}$^GFP, we assembled a 422 bp 5′-upstream fragment of the genomic VPS13 locus, VPS13$^{\Delta C}$^GFP, containing an internal GFP tag inserted after residue 499 (Li et al., 2020) and lacking the C-terminal 647 aa (aa 1–2,498), and a 501 bp 3′-downstream fragment of the genomic VPS13 locus in a pRS423 plasmid backbone using gap repair cloning.

S. cerevisiae cells were grown in liquid synthetic complete dextrose medium (SD, 2% α-D-glucose [Sigma-Aldrich],and 0.7% [wt/vol] yeast nitrogen base [BD Difco] complemented with defined amino acid composition) at 180 rpm and 30°C. Cells were grown to early log-phase, carefully washed five times, and resuspended in SD-N medium (2% [wt/vol] α-D-glucose [Sigma-Aldrich] and 0.17% [wt/vol] yeast nitrogen base without amino acids and ammonium sulfate [BD Difco]) for starvation at 180 rpm and 30°C.

To generate giant Ape1 oligomers, cells harboring a pRS425-prCUP1-APE1-prAPE1-tagBFP-APE1 plasmid were grown to very early log-phase and treated with SD medium containing 500 μM CuSO$_4$ (Roth) for 4 h at 180 rpm and 30°C until log-phase. Cells were washed five times and resuspended in SD-N medium for starvation (1 h) at 180 rpm and 30°C and analyzed by fluorescence microscopy.

### Fluorescence microscopy
S. cerevisiae cells were imaged in indicated media at room temperature in 96-well glass-bottom microplates (Greiner Bio-One). Images and semi-three-dimensional time-lapse images were acquired using a Dragonfly 500 spinning disk microscope (Andor) attached to an inverted Ti2 microscope stand (Nikon) with a CFI Plan Apo Lambda 60×/1.4 oil immersion objective (Nikon) and a Zyla 4.2 sCMOS camera (Andor). Fluorophores were excited with excitation lasers 405, 488, and 561 nm; emission was collected using 450/50, 525/50, and 600/50 nm bandpass filters. Image processing was performed with Fiji version 2.1.0 (Schindelin et al., 2012).

### Analysis of phagophore dynamics
WT, Δvps13, and vps13$^{\Delta PLT}$ cells expressing 2GFP-Atg8 and containing pRS425-prCUP1-APE1-prAPE1-tagBFP-APE1 were grown overnight to early log-phase (∼0.2 OD$_{600}$). We added 250 μM CuSO$_4$ (Roth) and incubated for 4 h at 180 rpm and 30°C. Cells were harvested, washed five times with, and resuspended in SD-N medium. After 1 h at 180 rpm and 30°C, cells were imaged by fluorescence microscopy 45 times every 2 min. Phagophore dynamics were analyzed as described in Schütter et al. (2020). In brief, the duration from the emergence of Atg8 puncta, until phagophores reached their maximum size on giant preApe1 oligomers, was recorded. To calculate phagophore size, phagophores were treated as spherical caps on spherical preApe1 oligomers. The diameters of giant preApe1 oligomers (2r) and Atg8 structures/caps (2a) were measured at the point of maximal size (t$_{max}$) using the line diagram tool in ImageJ (Version 2.1.0). The phagophore surface was obtained via first calculating cap height (h = r − (r$^2$ − a$^2$)$^{1/2}$) and then the surface area (A = 2πhr) for t$_{max}$.

### Analysis of autophagosome size and number
Autophagosome size was determined in Fiji version 2.1.0. A line diagram was drawn through the middle of an AP in the focal section of Z-stack images. A second line diagram was drawn right next to the analyzed AP to determine background signal intensities. Background signals were averaged and multiplied by a factor of 2.5 to ensure stringent size measurements compensating for heterogeneous background signals. AP size was determined by intensity values above the background of the AP line diagram with a step size of 0.1 μm.

Autophagosome numbers were determined by using the "multi point" tool of Fiji to randomly mark 50 cells. For each cell, the number of Atg8-positive puncta and autophagosomes defined as detectable ring structures was recorded. All experiments were analyzed blinded.

### Whole-cell extraction, Western blot analysis, and quantification
0.25 OD$_{600}$-units of yeast cells were harvested and lysed with 0.255 M NaOH (Roth). Proteins were precipitated in 50% (wt/vol) trichloroacetic acid (Roth) and washed once with ice-cold acetone (Merck). Protein pellets were resuspended in 1 × sodium dodecyl sulfate (SDS) sample buffer (50 mM Tris/HCl pH 6.8 [Roth], 10% [vol/vol] glycerol [Sigma-Aldrich], 1% [wt/vol] SDS [Roth], 0.01% [wt/vol] bromphenol blue [Roth], and 1% [vol/vol] β-mercaptoethanol [Merck]).

Proteins were analyzed by SDS-PAGE using primary antibodies in 5% (wt/vol) milk powder in TBST-T (monoclonal α-GFP [632381; Takara], polyclonal α-mCherry [GTX128508-100; GeneTex]; polyclonal α-Vps13 [Park et al., 2021]; and monoclonal α-Pgk1 [ab113687; Abcam]). Primary antibodies were visualized using secondary Dylight 800 α-mouse (610-145-002; Rockland Immunochemicals) or Dylight 800 α-rabbit (611-145-122; Rockland Immunochemicals) antibodies. The Li-COR Odyssey Infrared Imaging system (Biosciences) was used for detection and analysis of fluorescence signals in combination with the ImageStudoLite (Version 5.2.5) software.

For the calculation of autophagic flux, the signal intensity of free GFP bands was divided by the sum of the signal intensity of the total GFP signals comprised of 2GFP-Atg8 and free GFP. Protein expression levels of proteins tagged with mCherry were normalized to Pgk1.

### A quantitative model for autophagosome biogenesis
To approximate the number of phospholipids within the double-membranes of autophagosomes, we followed the calculations according to Melia et al. (2020) with an intermembrane distance

of 5 nm between the outer and inner membrane (Bieber et al., 2022). In short, for the data shown in Fig. 1 B, the surfaces of the autophagosomal outer and inner membranes were determined with $S_{outer} = 4\pi r^2 + 4\pi(r-5)^2$ and $S_{inner} = 4\pi(r-10)^2 + 4\pi(r-15)^2$ with $r$ = radius (1/2 diameter) of the autophagosome. The total surface area $S_{total} = S_{outer} + S_{inner}$ was converted into number of phospholipids assuming that 65 nm² of the surface area contains the equivalent of 100 molecules of phosphatidylcholine (Melia et al., 2020). The intermembrane volume of autophagosomes was calculated as $V_{intermembrane} = \frac{4}{3}\pi(r-5)^3 - \frac{4}{3}\pi(r-10)^3$. For the data shown in Fig. 1 C, we assumed that the intermembrane volume is exclusively derived from small Atg9 (60 nm) and COPII (80 nm) vesicles with the corresponding volumes of $V_{vesicles} = \frac{4}{3}\pi r^3$. Based on Yamamoto et al. (2012), we used three Atg9 vesicles and added the required number of COPII vesicles to match the calculated intermembrane volume of the autophagosomes. We then calculated the total surface area of the corresponding number of Atg9 and COPII vesicles as $S_{vesicles} = S_{Atg9} + S_{COPII}$ with $S_{Atg9} = 3(4\pi 30^2 + 4\pi(25)^2)$ and $S_{COPII} = n(4\pi 40^2 + 4\pi(35)^2)$. We assumed phospholipids/surface area not derived from vesicles must stem from non-vesicular PLT. Thus, to determine the contribution of non-vesicular PLT, we calculated 100% * $(S_{total} - S_{vesicles})/S_{total}$.

## Statistical analysis

Statistical analysis of all experiments from this study were performed in GraphPad Prism software 7.03. The number of independent experiments is stated in the figure legends. P values were calculated with two-tailed, unpaired $t$ test when comparing two sets of data or with two-way ANOVA when comparing more than two sets of data. Statistical significances are *, $P < 0.0332$; **, $P < 0.0021$; ***, $P < 0.0002$. Only significant changes are indicated in the graphs. Standard deviations are represented as error bars in the graphs.

## Bioinformatics

*S. cerevisiae* open reading frame (ORF) sequences were retrieved from the yeast genome database (yeastgenome.org) and used for plasmid maps and genomic modifications with the program SnapGene Version 4.3.11. Images were processed with Fiji Version 2.1.0, and figures and graphs were created with GraphPad Prism as well as Adobe Illustrator Version 24.1.3.

## Online supplemental material

Fig. S1 shows that Atg2-mediated PLT is essential for autophagosome formation. Fig. S2 shows an analysis of autophagy in the absence of Vps13 or known Vps13-adaptor proteins. Fig. S3 shows deficient Vps13-mediated PLT does not affect autophagy capacity.

## Data availability

The data underlying Figs. 1, 2, 3, 4, and 5; and Figs. S1, S2, and S3 are available in the published article and its online supplemental material.

## Acknowledgments

We would like to thank Thomas Langer for generous support; Charlotte Moter for excellent technical support and all members of the Graef lab for discussion; the FACS & Imaging Core Facility at the Max Planck Institute for Biology of Ageing for excellent support; and Karin Reinisch (Yale School of Medicine, Department of Cell Biology, New Haven, CT, USA) for providing the pRS423-Vps13^GFP plasmids.

This work was supported by the Max Planck Society and the Deutsche Forschungsgemeinschaft (DFG, German Research Foundation)—SFB 1218—project number 269925409 to M. Graef. Open Access funding provided by the Max Planck Society.

Author contributions: R. Dabrowski and M. Graef were the lead contributors to the conception, design, and interpretation of the experiments. R. Dabrowski performed all experiments and analyzed the data. S. Tulli supported data acquisition, analysis, and interpretation. R. Dabrowski and M. Graef wrote the manuscript.

Disclosures: The authors declare no competing interests exist.

Submitted: 10 November 2022

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

# Supplemental material

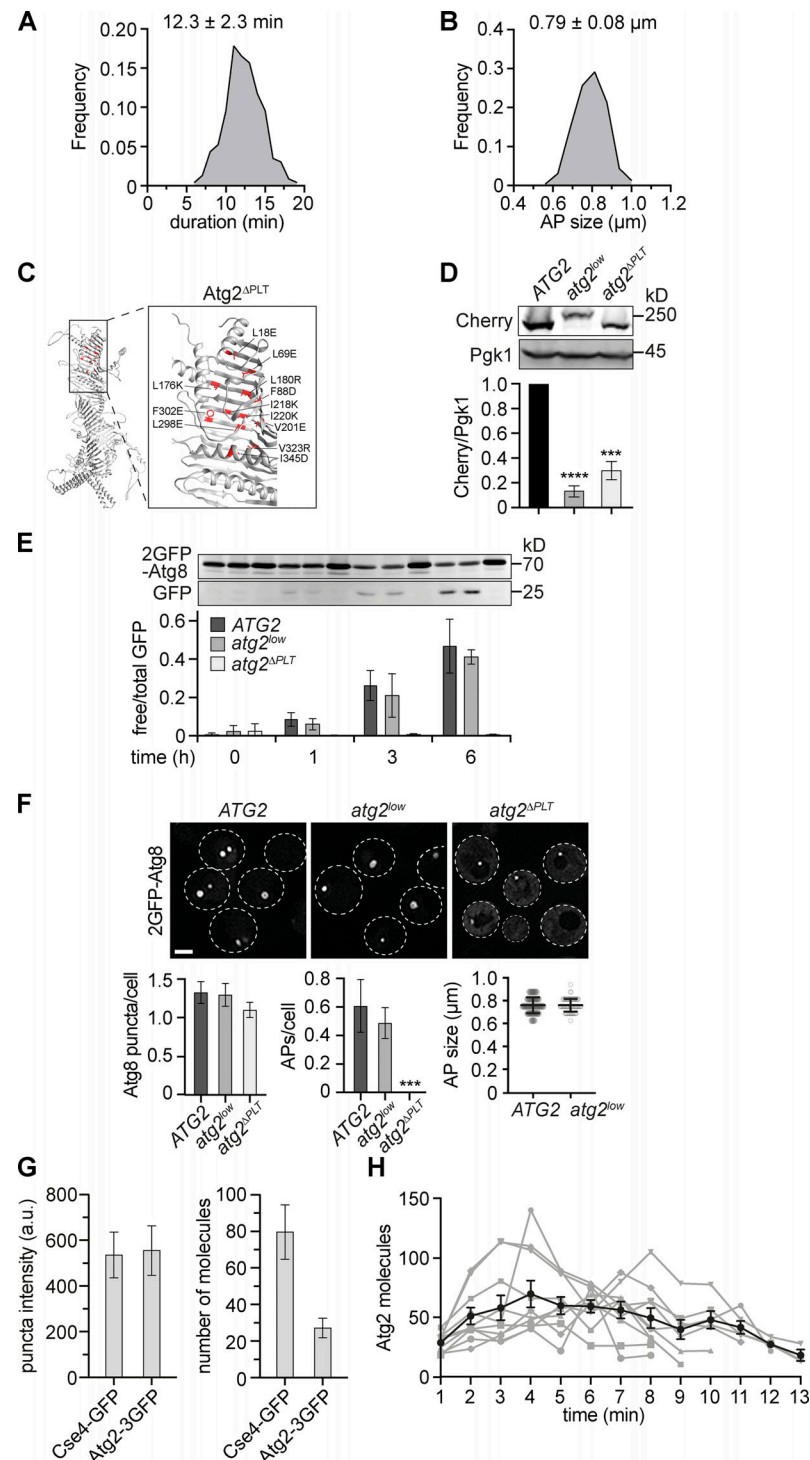

Figure S1. **Atg2-mediated PLT is essential for autophagosome formation. (A)** Relative frequency of autophagosome biogenesis durations from data shown in Fig. 1, D–F. **(B)** Relative frequency of autophagosome sizes from data shown in Fig. 1, D–F. **(C)** AlphaFold-based structure-prediction of *S. cerevisiae* Atg2 with mutated amino acid residues in Atg2$^{\Delta PLT}$ shown in red (Jumper et al., 2021). **(D)** Western blot analysis of whole cell extracts and quantifications of protein levels of Atg2-Cherry, Atg2$^{low}$-2Cherry, and Atg2$^{\Delta PLT}$-Cherry expressed from the endogenous *ATG2* locus using α-Cherry and α-Pgk1 antibodies. **(E)** Indicated strains expressing *2GFP-ATG8* were starved, and autophagy flux was analyzed at indicated time points by whole-cell extraction and Western blot analysis using an α-GFP antibody. Data are means ± SD (number of independent experiments, *n* = 4). **(F)** Fluorescence imaging of indicated strains expressing *2GFP-ATG8* after starvation (1 h). Quantification of the number of Atg8 puncta and autophagosomes (APs) per cell (number of independent experiments, *n* = 4; 200 cells/strain analyzed in total) and autophagosome size (*n* = 4; 40 APs/strain). Scale bar is 3 μm. **(G)** Fluorescence imaging of cells expressing *CSE4-GFP* or *ATG2-3GFP* after starvation (1 h). Fluorescence intensities for punctate signals were quantified, and the mean Cse4-GFP intensity was normalized to 80 molecules (number of independent experiments, *n* = 3; 150 cells/strain analyzed in total). **(H)** Number of Atg2 molecules during autophagosome biogenesis for 10 events in WT cells is shown in gray (included in Fig. 4 A). The mean ± SD over time for these 10 events is shown in black. Source data are available for this figure: SourceData FS1.

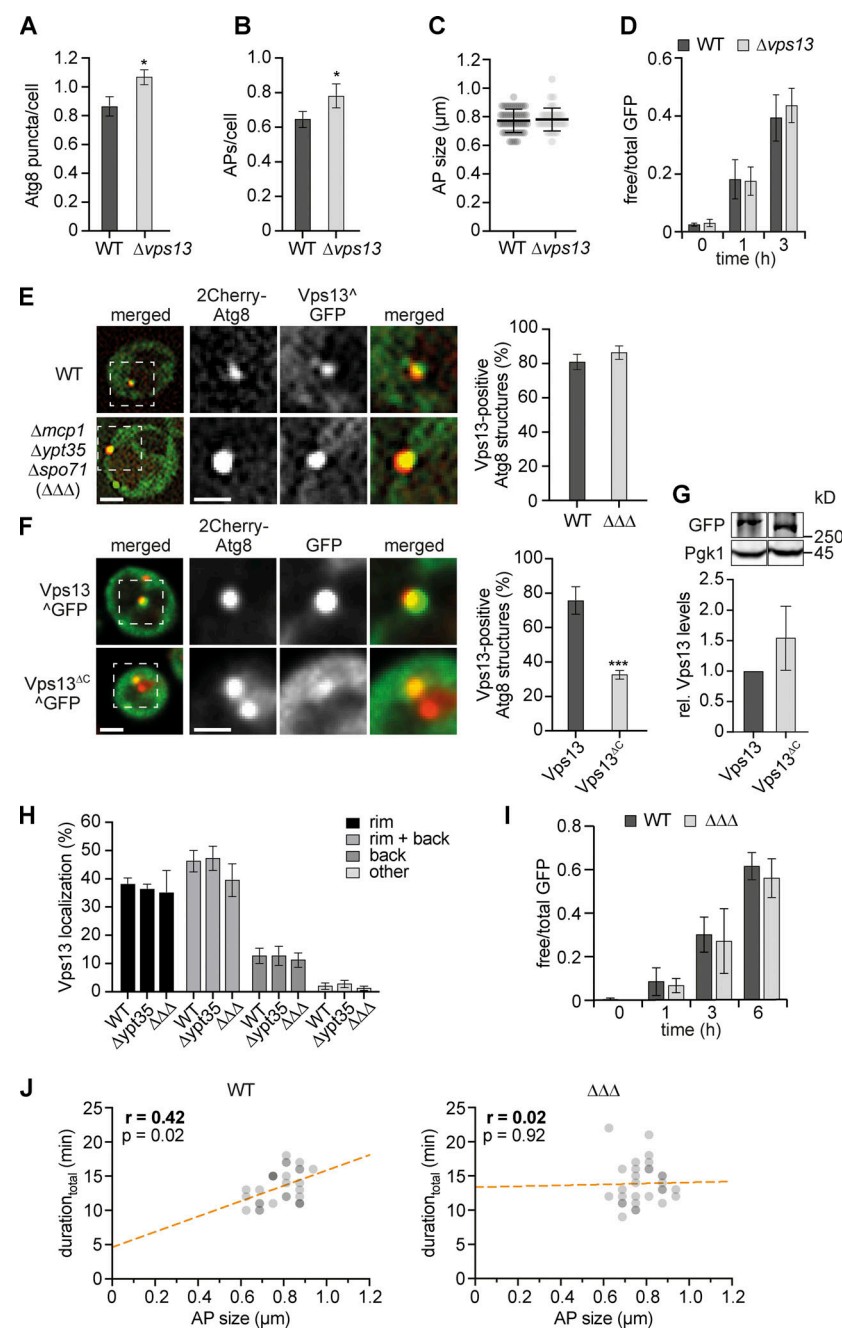

Figure S2. **Analysis of autophagy in the absence of Vps13 or known Vps13-adaptor proteins. (A–C)** Fluorescence imaging of WT and Δ*vps13* cells expressing *2GFP-ATG8* after starvation (1 h). **(A and B)** Quantification of the number of Atg8 puncta (A) and autophagosomes (APs; B; number of independent experiments, *n* = 4; 200 cells/strain analyzed in total). **(C)** Autophagosome size distribution for WT and Δ*vps13* cells (number of independent experiments, *n* = 4; 40 APs/strain analyzed in total). **(D)** Quantification of the autophagic flux of indicated strains expressing *2GFP-ATG8* during starvation. Data are means ± SD (number of independent experiments, *n* = 4). Cells were analyzed at indicated time points by whole cell extraction and Western blot analysis using α-GFP antibody. **(E)** Fluorescence imaging of WT and Δ*mcp1Δypt35Δspo71* (ΔΔΔ) cells expressing *2Cherry-ATG8* and pRS423-*VPS13^GFP* in a Δ*vps13* strain background after starvation (1 h). Quantification of Vps13-positive Atg8 structures shown in the right panel (number of independent experiments, *n* = 4; 200 structures/strain analyzed in total). Scale bars are 2 and 1 µm (zoom in). **(F)** Fluorescence imaging of cells expressing *2Cherry-ATG8* and either pRS423-*VPS13^GFP* or pRS423-*VPS13^GFP^ΔC* in Δ*vps13* background after starvation (1 h) and quantification of Vps13-positive Atg8 structures are shown in the right panel (number of independent experiments, *n* = 4; 200 structures/strain analyzed in total). Scale bars are 2 and 1 µm (zoom in). **(G)** Western blot analysis of whole cell extracts and quantifications of protein levels of Vps13^GFP and Vps13ΔC^GFP using α-Vps13 and α-Pgk1 antibodies. Data are means ± SD (number of independent experiments, *n* = 3). **(H)** Quantification of fluorescent images of Δ*vps13*, Δ*vps13 Δypt35*, and Δ*vps13 Δmcp1Δypt35Δspo71* (ΔΔΔ) cells expressing *2Cherry-ATG8*, pRS423-*VPS13^GFP*, and pRS425-*APE1-BFP* after starvation (1 h; number of independent experiments, *n* = 4; 200 structures analyzed in total). **(I)** Quantification of the autophagic flux of indicated strains expressing *2GFP-ATG8* during starvation. Data are means ± SD (number of independent experiments, *n* = 4). **(J)** Timelapse fluorescence imaging of WT and ΔΔΔ cells expressing *2GFP-ATG8* after starvation (1 h). Simple linear regression of duration$_{total}$ and autophagosome size in WT and ΔΔΔ cells after time-lapse fluorescence imaging of yeast cells expressing *2GFP-ATG8* after starvation (1 h; number of independent experiments, *n* = 3; 30 events/strain analyzed in total). WT data were included in Fig. 1, D–F. Source data are available for this figure: SourceData FS2.

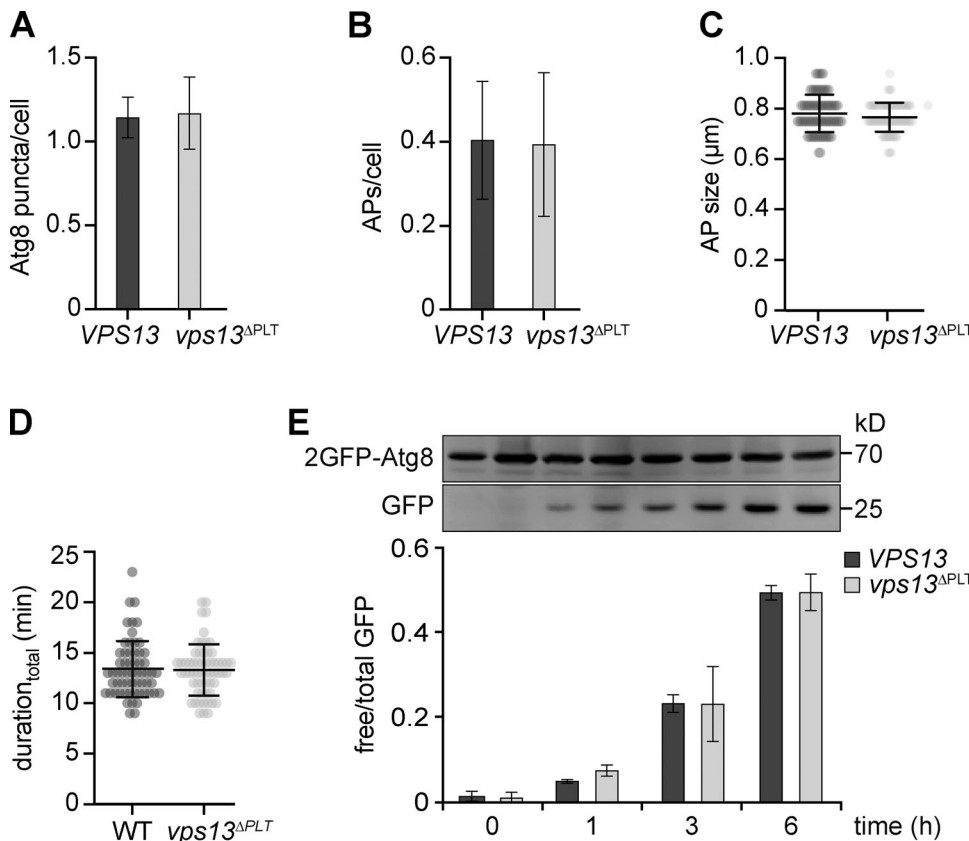

Figure S3. **Deficient Vps13-mediated PLT does not affect autophagy capacity. (A–D)** Fluorescence imaging of indicated strains expressing *2GFP-ATG8* after starvation (1 h). **(A and B)** Quantification of the number of Atg8 puncta (A) and autophagosomes (APs; B) per cell (number of independent experiments, $n = 4$; 200 cells/strain analyzed in total). **(C and D)** Autophagosome size distribution (C) and duration$_{total}$ of autophagosome biogenesis (D). Data are means ± SD (number of independent experiments, $n = 4$; 40 APs/strain analyzed in total). **(E)** Autophagic flux of indicated strains expressing *2GFP-ATG8* at indicated time points of starvation analyzed by whole-cell extraction and Western blot analysis using an α-GFP antibody. Data are means ± SD (number of independent experiments, $n = 4$). Source data are available for this figure: SourceData FS3.

