## [Peer Review File · The Journal of Cell Biology]

Parallel phospholipid transfer by Vps13 and Atg2 determines autophagosome biogenesis dynamics

Rahel Dabrowski, Susanna Tulli, and Martin Graef

Corresponding Author(s): Martin Graef, Cornell University

Review Timeline:

Submission Date:	2022-11-10
Editorial Decision:	2022-12-22
Revision Received:	2023-03-30
Editorial Decision:	2023-04-03
Revision Received:	2023-04-06

Monitoring Editor: Hong Zhang

Scientific Editor: Dan Simon

Transaction Report:

DOI: <https://doi.org/10.1083/jcb.202211039>

December 22, 2022

Re: JCB manuscript #202211039

Dr. Martin Graef
Cornell University
Molecular Biology and Genetics
107 Biotechnology Building 526 Campus Road
Ithaca, NY 14853

Dear Dr. Graef,

Thank you for submitting your manuscript entitled "Parallel phospholipid transfer by Vps13 and Atg2 determines autophagosome biogenesis dynamics." The manuscript was assessed by expert reviewers, whose comments are appended to this letter. We invite you to submit a revision if you can address the reviewers' key concerns, as outlined here.

You will see that overall the reviewers are enthusiastic about your study. Their comments request clarifications of the data and methods, further quantitative analyses to provide stronger supporting evidence for conclusions, and also additional insight into the dynamics of Vps13 accumulation on autophagosomes.

GENERAL GUIDELINES:

Text limits: Character count for a Report is < 20,000, not including spaces. Count includes title page, abstract, introduction, the joint Results & Discussion, and acknowledgments. Count does not include materials and methods, figure legends, references, tables, or supplemental legends.

Figures: Reports may have up to 5 main text figures. To avoid delays in production, figures must be prepared according to the policies outlined in our Instructions to Authors, under Data Presentation, <https://jcb.rupress.org/site/misc/ifora.xhtml>. All figures in accepted manuscripts will be screened prior to publication.

Supplemental information: There are strict limits on the allowable amount of supplemental data. Reports may have up to 3 supplemental figures. Up to 10 supplemental videos or flash animations are allowed. A summary of all supplemental material should appear at the end of the Materials and methods section.

Please note that JCB now requires authors to submit Source Data used to generate figures containing gels and Western blots with all revised manuscripts. This Source Data consists of fully uncropped and unprocessed images for each gel/blot displayed in the main and supplemental figures. Since your paper includes cropped gel and/or blot images, please be sure to provide one Source Data file for each figure that contains gels and/or blots along with your revised manuscript files. File names for Source Data figures should be alphanumeric without any spaces or special characters (i.e., SourceDataF#, where F# refers to the associated main figure number or SourceDataFS# for those associated with Supplementary figures). The lanes of the gels/blots should be labeled as they are in the associated figure, the place where cropping was applied should be marked (with a box), and molecular weight/size standards should be labeled wherever possible. Source Data files will be made available to reviewers during evaluation of revised manuscripts and, if your paper is eventually published in JCB, the files will be directly linked to specific figures in the published article.

The typical timeframe for revisions is three to four months. While most universities and institutes have reopened labs and allowed researchers to begin working at nearly pre-pandemic levels, we at JCB realize that the lingering effects of the COVID-19 pandemic may still be impacting some aspects of your work, including the acquisition of equipment and reagents. Therefore, if you anticipate any difficulties in meeting this aforementioned revision time limit, please contact us and we can work with you to find an appropriate time frame for resubmission. Please note that papers are generally considered through only one revision cycle, so any revised manuscript will likely be either accepted or rejected.

Thank you for this interesting contribution to Journal of Cell Biology. You can contact us at the journal office with any questions, cellbio@rockefeller.edu or call (212) 327-8588.

Sincerely,

Hong Zhang, PhD
Monitoring Editor
Journal of Cell Biology

Dan Simon, PhD
Scientific Editor
Journal of Cell Biology

Reviewer #1 (Comments to the Authors (Required)):

Several recent papers have suggested that lipid transport through Atg2 proteins may be a principal mechanism of autophagosome membrane expansion. Here, Graef and colleagues establish several key elements of Atg2 function to develop and set limits on a lipid-transfer model of membrane expansion. Their work establishes when and how many copies of Atg2 are present at the expanding phagophore as a function of its size. They build on previous work implicating a non-essential but detectable role for the lipid transport protein Vps13 and determine its localization during phagophore growth. Most intriguing, they compare Atg2 copy numbers with the ultimate size of the autophagosome and conclude there is very little correlation between the two suggesting that Atg2 numbers and Atg2-mediated lipid transport, are not rate-limiting for autophagosome biogenesis. However, deletion of Vps13 strongly increases the apparent correlation between Atg2 number and autophagosome size. Their work suggests that Vps13 and Atg2 function semi-redundantly to deliver lipids into the expanding autophagosome and when present together, ensure that lipid transfer alone is not rate-limiting in the biogenesis event.

Overall, this is a beautiful paper, the work is rigorous and the results will certainly influence models of autophagosome biogenesis moving forward. My only significant question relates to whether, as designed, the experiments test which steps are rate-limiting specifically for autophagosome biogenesis. The authors look at whether the final size of the autophagosome correlates with the total time of autophagosome biogenesis which they define as spanning the earliest detection of Atg8 to the ultimate loss of GFP-Atg8 as its consumed in the vacuole. I am concerned that the events following growth that lead to vacuole consumption, including closure of the autophagosome and fusion in the vacuole, may well be rate-limiting in most cases, but should not be considered part of the "autophagosome biogenesis" pathway. If the proposed role of Atg2 is to drive lipids in support of membrane expansion specifically, it seems like it would be better to correlate size versus time specifically over the periods where the phagophore is undergoing rapid expansion, which may be only the first 4 or 5 minutes in the images shown.

Minor points:

- 1) Related to point above, there are many places in the text where the authors relate their duration experiments to biogenesis. For example, when plotting autophagosome maximum diameter against duration, "Taken together, these data indicate that autophagic membrane assembly, critically driven by phospholipid synthesis, transfer, and scrambling, is not a generally rate-limiting factor for autophagosome formation." Not sure if duration in this case is really of autophagosome formation so much as autophagosome lifetime. Ideally, plots of Atg2 accumulation over time versus autophagosome size could be used to better explore the dynamics during periods where the size is clearly changing and thus is still upstream of closure/docking/fusion steps. However, it may also suffice to discuss this potential caveat.
- 2) The authors do a great job of tracking Atg2 accumulation at the phagophore over time. Can they do the same for Vps13? Are Vps13 proteins in place during the rapid expansion of the phagophore that is apparent in the early time points of their data and related, is there a temporal explanation for the ~20% of Atg2 puncta that are Vps13 negative. One could imagine a need for a specific lipid in support of a late event such as closure (as suggested by the recent Elazar paper) or in support of fusion to the vacuole.
- 3) Does loss of the other Vps13 adaptors change the distribution of Vps13 on Giant Ape1 phagophores? In particular, is the localization to the backside, presumably adjacent to the vacuole, dependent on known vacuole targeting adaptor (ypt35)?

Reviewer #2 (Comments to the Authors (Required)):

Dabrowski et al in this short report examine the activity and requirement for Atg2 in autophagosome biogenesis. Using quantitative fluorescence imaging they determine several key aspects of the phospholipid transfer protein Atg2, known to be essential for autophagy. In agreement with previous data, Atg2 is required to expand the phagophore but for efficient and sufficient autophagy the activity of Vps13 is essential.

The approach is elegant and the data compelling. There are a few points to address.

1. In Figure S1B, it is not clear what is integrated in the ATG2 strain, just a cherry tag or was the whole gene replaced with a cherry tag. There is a difference in the molecular weights of the cherry-Atg2 constructs. Why wasn't a WT (un-edited) strain included in Figure 1SC-D? especially given the concern about protein expression levels?
2. Figure 1H why the use of Cherry-Atg8 and GFP-ATG2 and not GFP-Atg8 and "low" cherry-Atg2? Does the C-terminally triple tagged Atg2 rescue the deltaAtg2 deletion mutant?
3. Figure 2A there is no information about how the structure of the Rat VPS13A was produced. There is no citation to alphafold or other computational methods.
4. Bottom of page 10, the authors describe the size of autophagosomes being around 100µm in the absence of Vps13 at the intersection of the x-axis. However the scale in Figure 3C would suggest the value would be less than 0.1 µm.
5. 3rd line from bottom of page 10, of not or.
6. The authors unexpectedly mention Atg9 and COPII vesicles at the top of page 11, and don't include these in the description of Figure 4E. It might be helpful to non-experts to describe the model more completely in the conclusion.

Reviewer #3 (Comments to the Authors (Required)):

The manuscript by Dabrowski et al sets out to solve a conundrum during autophagosome biogenesis in yeast. When measuring the duration of ATG8 puncta vs their size the authors obtained a range of relations away from linear. To explain this discrepancy the authors hypothesised that autophagic membrane assembly (driven by phospholipid supply) is not rate limiting for autophagosome formation. When looking at candidates for lipid supply, they show that the maximum number of ATG2 molecules (the main protein supplying lipid to the forming phagophore) does not correlate with the duration or size of the forming autophagosomes reinforcing the idea that the ATG2-mediated lipid supply is not limiting.

Further work shows that another lipid supplying protein, VPS13, although not essential for autophagy it appears to be able to also supply lipids to the forming phagophore since its absence makes the relation between ATG8 puncta duration and size more linear. The mechanism by which VPS13 engages with the phagophore is not dependent on its known interacting proteins for other lipid supply pathways.

I like what this paper is trying to do but I am not convinced that it has a strong conclusion at this point.

One issue is that the data of the puncta measurements are noisy so I am not sure that linearity is not hidden somewhere there to be obtained. Incidentally, it will be good to show a linear plot of the durations in Fig 1I instead of this discontinuous format. Another issue is whether we actually expect a linear relationship based on the fact that the autophagosome (spherical-like) diameter will increase roughly with the square root of the area. Have the authors done a regression between duration and size-square?

Finally, the implied function of VPS13 in this pathway would need to be further strengthened by finding out how does it get to the autophagosomes? In addition, because the protein seems to be in many cellular locations with only a few puncta colocalizing with forming autophagosomes, it will be good to see the dynamics of this during rounds of expansion and collapse. Do the authors see one discreet VPS13 structure co-localizing with a forming autophagosome throughout, or do they see several VPS13 structures coming on and off?

Minor: Apologies if it is stated somewhere in the text, but I could not see an acquisition rate for the movies shown. For this type of analysis, this rate ought to be as high as possible without introducing photo-toxicity.

We would like to thank the reviewers for their time, the very positive assessment of our work and the fair and constructive comments and suggestions. We have now revised the manuscript and included the requested experiments and changes. Please find our point-by-point discussion below:

Reviewer #1 (Comments to the Authors (Required)):

Several recent papers have suggested that lipid transport through Atg2 proteins may be a principal mechanism of autophagosome membrane expansion. Here, Graef and colleagues establish several key elements of Atg2 function to develop and set limits on a lipid-transfer model of membrane expansion. Their work establishes when and how many copies of Atg2 are present at the expanding phagophore as a function of its size. They build on previous work implicating a non-essential but detectable role for the lipid transport protein Vps13 and determine its localization during phagophore growth. Most intriguing, they compare Atg2 copy numbers with the ultimate size of the autophagosome and conclude there is very little correlation between the two suggesting that Atg2 numbers and Atg2-mediated lipid transport, are not rate-limiting for autophagosome biogenesis. However, deletion of Vps13 strongly increases the apparent correlation between Atg2 number and autophagosome size. Their work suggests that Vps13 and Atg2 function semi-redundantly to deliver lipids into the expanding autophagosome and when present together, ensure that lipid transfer alone is not rate-limiting in the biogenesis event.

Overall, this is a beautiful paper, the work is rigorous and the results will certainly influence models of autophagosome biogenesis moving forward. My only significant question relates to whether, as designed, the experiments test which steps are rate-limiting specifically for autophagosome biogenesis. The authors look at whether the final size of the autophagosome correlates with the total time of autophagosome biogenesis which they define as spanning the earliest detection of Atg8 to the ultimate loss of GFP-Atg8 as its consumed in the vacuole. I am concerned that the events following growth that lead to vacuole consumption, including closure of the autophagosome and fusion in the vacuole, may well be rate-limiting in most cases, but should not be considered part of the "autophagosome biogenesis" pathway. If the proposed role of Atg2 is to drive lipids in support of membrane expansion specifically, it seems like it would be better to correlate size versus time specifically over the periods where the phagophore is undergoing rapid expansion, which may be only the first 4 or 5 minutes in the images shown.

Minor points:

1) Related to point above, there are many places in the text where the authors relate their duration experiments to biogenesis. For example, when plotting autophagosome maximum diameter against duration, "Taken together, these data indicate that autophagic membrane assembly, critically driven by phospholipid synthesis, transfer, and scrambling, is not a generally rate-limiting factor for autophagosome formation." Not sure if duration in this case is really of autophagosome formation so much as autophagosome lifetime. Ideally, plots of Atg2 accumulation over time versus autophagosome size could be used to better explore the dynamics during periods where the size is clearly changing and thus is still upstream of closure/docking/fusion steps. However, it may also suffice to discuss this potential caveat.

We have carefully addressed this well-taken point and included the duration until the phagophore/autophagosome reaches its maximal size/diameter (defined as $\text{duration}_{\text{max}}$). We anticipate this timeframe to be determined by the expansion rate of the nucleated phagophore. Consistent with our prior conclusions, we found the same relationships between $\text{duration}_{\text{max}}$ and autophagosome size as for the total duration ($\text{duration}_{\text{total}}$). Both $\text{duration}_{\text{max}}$ and $\text{duration}_{\text{total}}$ are now included in the manuscript in **new figures 1E, 3A-G, and 5C-G**. In addition, we measured phagophore growth on enlarged Ape1 oligomers similar to our analyses described in Schuetter et al. Cell 2020. Importantly, we also observed a clearly improved positive correlation between the duration and maximal phagophore size in this assay in the absence of Vps13 compared with WT cells (**new figure 3H-J**). Taken together, our previous and new data strongly support our model that Atg2-mediated phospholipid transfer (PLT) becomes rate-limiting for autophagosome biogenesis in the absence of parallel PLT by Vps13.

2) The authors do a great job of tracking Atg2 accumulation at the phagophore over time. Can they do the same for Vps13? Are Vps13 proteins in place during the rapid expansion of the phagophore that is apparent

in the early time points of their data and related, is there a temporal explanation for the ~20% of Atg2 (Atg8) puncta that are Vps13 negative. One could imagine a need for a specific lipid in support of a late event such as closure (as suggested by the recent Elazar paper) or in support of fusion to the vacuole.

We added the timelapse analysis monitoring autophagosome biogenesis (Atg8) and endogenous Vps13^ΔGFP in the new **figure 2C**. Similar to Atg2, we find Vps13 puncta consistently linked to forming autophagosomes. Unfortunately, at endogenous protein levels, we do not obtain sufficient data to perform quantitative analyses for Vps13 similar to Atg2.

In addition, we have included the analysis of Atg2 puncta at autophagic structures in the new **figure 2B** for comparison with Vps13. We find roughly 20% of Atg8-marked structures without Atg2 or Vps13, respectively. Given that both, Atg2 and Vps13 dissociated from mature autophagosomes in our timelapse analysis, we consider this to be the main reason for Atg2- or Vps13-negative Atg8 structures. In addition, we cannot exclude that signals below our detection limit may contribute.

3) Does loss of the other Vps13 adaptors change the distribution of Vps13 on Giant Ape1 phagophores? In particular, is the localization to the backside, presumably adjacent to the vacuole, dependent on known vacuole targeting adaptor (ypt35)?

Interestingly, the deletion of ypt35 or of all three known adaptors of Vps13 did not affect the localization of Vps13 to the rim or the back side/vacuole (new **figure S2H**). In response to reviewer 3, we found that the C-terminal sequence of Vps13 is required for efficient localization of Vps13 to forming autophagosomes (new **figure S2F**). Based on our data, we hypothesize that as yet unknown adaptor(s) likely exist for recruiting Vps13 specifically to forming autophagosomes.

Reviewer #2 (Comments to the Authors (Required)):

Dabrowski et al in this short report examine the activity and requirement for Atg2 in autophagosome biogenesis. Using quantitative fluorescence imaging they determine several key aspects of the phospholipid transfer protein Atg2, known to be essential for autophagy. In agreement with previous data, Atg2 is required to expand the phagophore but for efficient and sufficient autophagy the activity of Vps13 is essential.

The approach is elegant and the data compelling. There are a few points to address.

1. In Figure S1B, it is not clear what is integrated in the ATG2 strain, just a cherry tag or was the whole gene replaced with a cherry tag. There is a difference in the molecular weights of the cherry-Atg2 constructs. Why wasn't a WT (un-edited) strain included in Figure 1SC-D? especially given the concern about protein expression levels?

We introduced either the WT (ATG2) or the PLT-deficient variant Atg2^ΔPLT into the genomic locus of ATG2 as C-terminally mCherry-tagged variants. The atg2^{low} variant carries two C-terminal mCherry tags, which cause the slower mobility in the WB analysis. The measurement of protein abundance takes the mCherry copy number into account. We added more explanation to the figure legend of S1D.

2. Figure 1H why the use of Cherry-Atg8 and GFP-ATG2 and not GFP-Atg8 and "low" cherry-Atg2? Does the C-terminally triple tagged Atg2 rescue the deltaAtg2 deletion mutant?

Atg2 puncta contain a low protein copy number and are difficult to image especially over longer time. Thus, we prefer the 3GFP-tagged Atg2 variant for timelapse imaging. It fully rescues the phenotype of a ATG2 deletion and is functional.

3. Figure 2A there is no information about how the structure of the Rat VPS13A was produced. There is no citation to AlphaFold or other computational methods.

We cite the paper by Jumper et al. 2021 describing AlphaFold in the figure legend.

4. Bottom of page 10, the authors describe the size of autophagosomes being around 100 μ m in the absence of Vps13 at the intersection of the x-axis. However the scale in Figure 3C would suggest the value would be less than 0.1 μ m.

Thank you for pointing this out. We apologize for the mistake. The text should have said "100 nm". We corrected the mistake in the revised version.

5. 3rd line from bottom of page 10, of not or.

Thank you. We corrected the mistake.

6. The authors unexpectedly mention Atg9 and COPII vesicles at the top of page 11, and don't include these in the description of Figure 4E. It might be helpful to non-experts to describe the model more completely in the conclusion.

Thank you for pointing this out. We added a short description to the figure legend.

Reviewer #3 (Comments to the Authors (Required)):

The manuscript by Dabrowski et al sets out to solve a conundrum during autophagosome biogenesis in yeast. When measuring the duration of ATG8 puncta vs their size the authors obtained a range of relations away from linear. To explain this discrepancy the authors hypothesised that autophagic membrane assembly (driven by phospholipid supply) is not rate limiting for autophagosome formation. When looking at candidates for lipid supply, they show that the maximum number of ATG2 molecules (the main protein supplying lipid to the forming phagophore) does not correlate with the duration or size of the forming autophagosomes reinforcing the idea that the ATG2-mediated lipid supply is not limiting. Further work shows that another lipid supplying protein, VPS13, although not essential for autophagy it appears to be able to also supply lipids to the forming phagophore since its absence makes the relation between ATG8 puncta duration and size more linear. The mechanism by which VPS13 engages with the phagophore is not dependent on its known interacting proteins for other lipid supply pathways.

I like what this paper is trying to do but I am not convinced that it has a strong conclusion at this point. One issue is that the data of the puncta measurements are noisy so I am not sure that linearity is not hidden somewhere there to be obtained.

Biological systems produce inherent biochemical noise. In addition to noise, cells may display heterogeneity in terms of autophagosome biogenesis. While we cannot exclude that a certain level of noise is introduced by our measurements, we expect a considerable level of noise/heterogeneity is biological in nature. Regardless of the origin and extent of potential noise, the validity of our approach is supported by the fact that we do detect improved linearity with our measurements in Vps13-deficient cells.

Incidentally, it will be good to show a linear plot of the durations in Fig 1I instead of this discontinuous format.

We added 10 examples of individual biogenesis events to the revised manuscript in new **figure S1H**. The data exemplifies the heterogeneity in the expansion dynamics of forming autophagosomes and the associated number of Atg2 molecules. In order to compare the dynamics of Atg2 assembly, we defined the three stages as shown in now **figure 1H**.

Another issue is whether we actually expect a linear relationship based on the fact that the autophagosome (spherical-like) diameter will increase roughly with the square root of the area. Have the authors done a regression between duration and size-square?

This is absolutely correct. Interestingly, within the observed size range of autophagosomes in vivo, the diameter correlates in a linear manner ($r=0.995$) with the surface area and number of phospholipids as shown in **figure 1B**. Based on this relationship, the regression between duration and the surface area shows the same behavior as in relation to the diameter. To clarify this point, we added a more direct explanation to the revised manuscript.

Finally, the implied function of VPS13 in this pathway would need to be further strengthened by finding out how does it get to the autophagosomes?

The reviewer raises an interesting question. We tested the role of the C-terminus of Vps13 for its recruitment to forming autophagosomes. We found that it plays a critical role as shown in new **figure S2F**. While we will characterize the molecular details of Vps13 binding to autophagic membranes, this will likely involve coincidence binding of lipids and proteins. Given that none of the known adaptors play a significant role, we will need to identify so far unknown autophagy-specific interactors of Vps13. However, to address this point comprehensively, we would require substantially more time and work. Whatever the potential mechanisms, we do not think it likely that their discovery will change any of our major conclusion in the present manuscript. For all these stated reasons, we consider comprehensively addressing this point is beyond the scope of the current manuscript.

In addition, because the protein seems to be in many cellular locations with only a few puncta colocalizing with forming autophagosomes, it will be good to see the dynamics of this during rounds of expansion and collapse. Do the authors see one discreet VPS13 structure co-localizing with a forming autophagosome throughout, or do they see several VPS13 structures coming on and off?

To address this point, we performed timelapse analysis of yeast cells expressing Cherry-Atg8 and endogenously expressed Vps13²GFP shown in new **figure 2C**. We observed Vps13 puncta that are likely continuously linked to forming autophagosomes similar to Atg2. With the limitations of our temporal resolution, we think Vps13 forms stable contact sites with the expanding phagophore.

Minor: Apologies if it is stated somewhere in the text, but I could not see an acquisition rate for the movies shown. For this type of analysis, this rate ought to be as high as possible without introducing photo-toxicity.

The acquisition rates in minutes are indicated in the figures.

April 3, 2023

RE: JCB Manuscript #202211039R

Prof. Martin Graef
Cornell University
Molecular Biology and Genetics
201 Biotechnology Building 526 Campus Road
Ithaca, NY 14853

Dear Prof. Graef,

Thank you for submitting your revised manuscript entitled "Parallel phospholipid transfer by Vps13 and Atg2 determines autophagosome biogenesis dynamics." We would be happy to publish your paper in JCB pending final revisions necessary to meet our formatting guidelines (see details below).

A. MANUSCRIPT ORGANIZATION AND FORMATTING:

1) Text limits: Character count for Reports is < 20,000, not including spaces. Count includes title page, abstract, introduction, results, discussion, and acknowledgments. Count does not include materials and methods, figure legends, references, tables, or supplemental legends.

2) Figure formatting: Reports may have up to 5 main text figures. Scale bars must be present on all microscopy images, including inset magnifications. Molecular weight or nucleic acid size markers must be included on all gel electrophoresis.

Also, please avoid pairing red and green for images and graphs to ensure legibility for color-blind readers. If red and green are paired for images, please ensure that the particular red and green hues used in micrographs are distinctive with any of the colorblind types. If not, please modify colors accordingly or provide separate images of the individual channels.

3) Statistical analysis: Error bars on graphic representations of numerical data must be clearly described in the figure legend. The number of independent data points (n) represented in a graph must be indicated in the legend. Please, indicate whether 'n' refers to technical or biological replicates (i.e. number of analyzed cells, samples or animals, number of independent experiments). If independent experiments with multiple biological replicates have been performed, we recommend using distribution-reproducibility SuperPlots (please see Lord et al., JCB 2020) to better display the distribution of the entire dataset, and report statistics (such as means, error bars, and P values) that address the reproducibility of the findings.

Statistical methods should be explained in full in the materials and methods. For figures presenting pooled data the statistical measure should be defined in the figure legends. Please also be sure to indicate the statistical tests used in each of your experiments (both in the figure legend itself and in a separate methods section) as well as the parameters of the test (for example, if you ran a t-test, please indicate if it was one- or two-sided, etc.). Also, if you used parametric tests, please indicate if the data distribution was tested for normality (and if so, how). If not, you must state something to the effect that "Data distribution was assumed to be normal but this was not formally tested."

4) Materials and methods: Should be comprehensive and not simply reference a previous publication for details on how an experiment was performed. Please provide full descriptions (at least in brief) in the text for readers who may not have access to referenced manuscripts. The text should not refer to methods "...as previously described." Please also indicate the type of membrane used for immunoblotting/western blots as well as acquisition and quantification methods for quantifications.

5) For all cell lines, vectors, constructs/cDNAs, etc. - all genetic material: please include database / vendor ID (e.g., Addgene, ATCC, etc.) or if unavailable, please briefly describe their basic genetic features, even if described in other published work or gifted to you by other investigators (and provide references where appropriate). Please be sure to provide the sequences for all of your oligos: primers, si/shRNA, RNAi, gRNAs, etc. in the materials and methods. You must also indicate in the methods the source, species, and catalog numbers/vendor identifiers for all of your antibodies, including secondary. If antibodies are not commercial, please provide a reference citation.

6) Microscope image acquisition: The following information must be provided about the acquisition and processing of images:

- a. Make and model of microscope
- b. Type, magnification, and numerical aperture of the objective lenses
- c. Temperature
- d. Imaging medium
- e. Fluorochromes
- f. Camera make and model
- g. Acquisition software
- h. Any software used for image processing subsequent to data acquisition. Please include details and types of operations involved (e.g., type of deconvolution, 3D reconstitutions, surface or volume rendering, gamma adjustments, etc.).

7) References: There is no limit to the number of references cited in a manuscript. References should be cited parenthetically in the text by author and year of publication. Abbreviate the names of journals according to PubMed.

8) Supplemental materials: Reports may have up to 5 supplemental figures and 10 videos. Please also note that tables, like figures, should be provided as individual, editable files. A summary of all supplemental material should appear at the end of the Materials and methods section. Please include one brief sentence per item.

9) eTOC summary: A ~40-50 word summary that describes the context and significance of the findings for a general readership should be included on the title page. The statement should be written in the present tense and refer to the work in the third person. It should begin with "First author name(s) et al..." to match our preferred style.

10) Conflict of interest statement: JCB requires inclusion of a statement in the acknowledgements regarding competing financial interests. If no competing financial interests exist, please include the following statement: "The authors declare no competing financial interests." If competing interests are declared, please follow your statement of these competing interests with the following statement: "The authors declare no further competing financial interests."

11) A separate author contribution section is required following the Acknowledgments in all research manuscripts. All authors should be mentioned and designated by their first and middle initials and full surnames. We encourage use of the CRediT nomenclature (<https://casrai.org/credit/>).

12) ORCID IDs: ORCID IDs are unique identifiers allowing researchers to create a record of their various scholarly contributions in a single place. At resubmission of your final files, please consider providing an ORCID ID for as many contributing authors as possible.

13) Please note that JCB now requires authors to submit Source Data used to generate figures containing gels and Western blots with all revised manuscripts. This Source Data consists of fully uncropped and unprocessed images for each gel/blot displayed in the main and supplemental figures. Since your paper includes cropped gel and/or blot images, please be sure to provide one Source Data file for each figure that contains gels and/or blots along with your revised manuscript files. File names for Source Data figures should be alphanumeric without any spaces or special characters (i.e., SourceDataF#, where F# refers to the associated main figure number or SourceDataFS# for those associated with Supplementary figures). The lanes of the gels/blots should be labeled as they are in the associated figure, the place where cropping was applied should be marked (with a box), and molecular weight/size standards should be labeled wherever possible. Source Data files will be directly linked to specific figures in the published article. Source Data Figures should be provided as individual PDF files (one file per figure). Authors should endeavor to retain a minimum resolution of 300 dpi or pixels per inch. Please review our instructions for export from Photoshop, Illustrator, and PowerPoint here: <https://rupress.org/jcb/pages/submission-guidelines#revised>

14) Journal of Cell Biology also requires a data availability statement for all research article submissions. These statements will be published in the article directly above the Acknowledgments. The statement should address all data underlying the research presented in the manuscript. Please visit the JCB instructions for authors for guidelines and examples of statements at (<https://rupress.org/jcb/pages/editorial-policies#data-availability-statement>).

B. FINAL FILES:

****It is JCB policy that if requested, original data images must be made available to the editors. Failure to provide original images upon request will result in unavoidable delays in publication. Please ensure that you have access to all original data images prior to final submission.****

****The license to publish form must be signed before your manuscript can be sent to production. A link to the electronic license to publish form will be sent to the corresponding author only. Please take a moment to check your funder requirements before choosing the appropriate license.****

Thank you for this very interesting contribution, we look forward to publishing your paper in Journal of Cell Biology.

Sincerely,

Hong Zhang, PhD
Monitoring Editor
Journal of Cell Biology

Dan Simon, PhD
Scientific Editor
Journal of Cell Biology